# FF-Erase : Machine Unlearning and Verification for Forward-Forward Models

## Abstract

The Forward-Forward (FF) algorithms present promising and biologically plausible alternatives to backpropagation (BP), enabling efficient model training through layer-wise greedy optimization. However, the critical task of machine unlearning for FF models, which involves efficiently removing specific training data's influence without full retraining, remains a foundational yet unexplored problem. The inherent characteristics of FF models, such as their sensitivity to parameter tuning and layer-wise independent training, pose unique challenges, often causing catastrophic model collapse when applying conventional unlearning methods. To fill this gap, we introduce a novel unlearning framework specifically for FF models, which employs a goodness-guided strategy. This method proposes a stable guidance model to generate target goodness distributions, steering the original model to unlearn forgetting data by shifting its layer-wise goodness scores, thereby effectively adapting gradient-based unlearning for the FF architecture. To enable robust verification on unlearning performance, we also propose a novel goodness-based membership inference attack (G-MIA), a powerful and lightweight black-box attack that leverages the unique properties of FF models' goodness scores. Our experiments demonstrate that our proposed method effectively removes the influence of target forgetting data on FF models while preserving model utility on the remaining data. Critically, our approach accomplishes 1.9 to $3.1\times$ faster than retraining from scratch, establishing an efficient foundation for FF unlearning.

## 1 Introduction

The Forward-Forward (FF) Hinton (2022) algorithms have emerged as a promising alternative to backpropagation (BP) for training deep learning models. This approach updates model parameters by greedily optimizing a layer-wise "goodness" score, which reflects the activation level of neurons in a layer. By maximizing this score for positive data (*i.e.*, valid training data with correct labels) and minimizing it for negative data (*e.g.*, invalid data or incorrectly labeled data) during forwarding, the FF algorithms effectively train model parameters without requiring a backward pass that blocks all layers. This BP-free nature is considered more biologically plausible and brings significant practical advantages, including reduced memory overhead from stored activations and the potential for efficient training using pipeline parallelism. These features make FF particularly well-suited for training on resource-constrained scenarios, such as in edge computing.

However, the critical task of machine unlearning for FF models remains a foundational yet unexplored problem. Usually, machine learning applications involve analyzing sensitive individuals' data. Their owners require the "right to be forgotten" (RTBF), which has been explicitly stated in the European Union General Data Protection Regulation (GDPR)Voigt & Von dem Bussche (2017) and the California Consumer Privacy Act (CCPA)Harding et al. (2019). Moreover, the model owners also need to remove outdated or poisoned data to promote model performanceWang et al. (2025b); Zhang et al. (2023). Machine unlearning achieves these data erasing goals by removing the influence of specific training samples from a trained model (*i.e.*, effectiveness) while preserving the model performance on the remaining data (*i.e.*, model utility).

Existing machine unlearning methods are not feasible for FF models. The most straightforward approach, retraining the model from scratch on the remaining data, is computationally prohibitive and impractical. Other unlearning methods calibrate the model parameters by either directly performing gradient ascent (GA) on the forgetting data Tarun et al. (2023a); Sekhari et al. (2021a) or estimating the parameters tuning Qiao et al. (2024); Liu et al. (2022b). As illustrated in Figure 1, they are also not applicable due to the unique challenges posed by the BP-free nature and layer-wise training of FF models. The specific details are as follows.

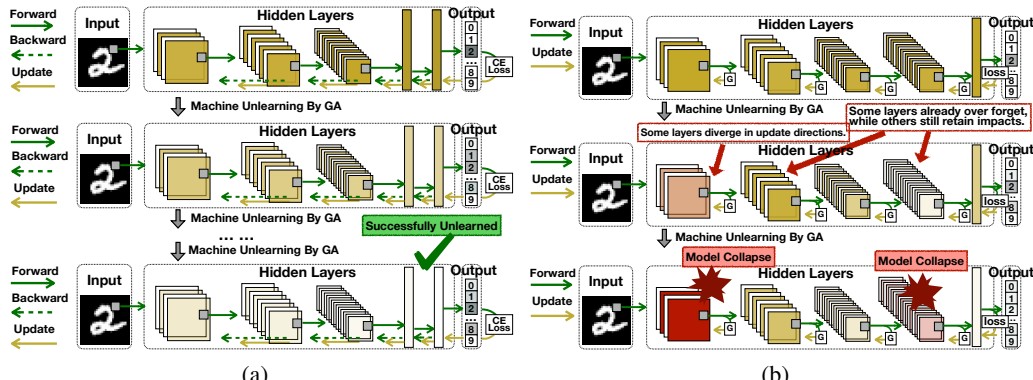

Figure 1: Classical machine unlearning methods (*e.g.*, gradient ascent) successfully adjust the loss and unlearn forgetting data on BP models (a). However, those methods result in model collapse and fail in FF models due to unique challenges as shown in illustration (b).

Firstly, FF models exhibit heightened sensitivity to parameter tuning due to their BP-free nature. BP methods utilize backpropagation to ensure consistent parameter update directions, thereby enhancing robustness to tuning variations. In contrast, FF algorithms use greedy and layer-wise training approaches, where each layer is independently optimized on its local goodness objective until the overall goodness scores converge to a specific distribution. In this process, the parameters in the previous layers do not strictly update towards a consistent direction with the subsequent layers, nor compress everything "useful" for the final output layer. Therefore, without careful design to prevent goodness from shifting to invalid distributions, layers may diverge in update directions during unlearning, risking model collapse. However, determining the validity of a goodness distribution in advance remains challenging, making it difficult to reliably guide layer updates during unlearning.

Secondly, the independent layer-wise training of FF models further complicates the unlearning process. In BP models, a common unlearning strategy is to perform gradient ascent on the loss function of the data to be removed, updating all layers jointly through the chain rule Gupta et al. (2021); Tarun et al. (2023b); Sekhari et al. (2021b); Chundawat et al. (2023b). In contrast, FF models optimize separate objective functions at each layer, with varying degrees of goodness improvement. This independence creates a key difficulty: it is unclear how much each layer's goodness should be penalized given a forgetting data sample. As a result, some layers may continue to over-forget while others only partially retain residual effects, thereby complicating the trade-off between effective unlearning and preserving the overall model utility.

The above discussion motivates us to answer the first key question: *How to design an efficient machine unlearning method for FF models to ensure both effectiveness and model utility?*

Moreover, it is also challenging to verify the effectiveness of an unlearning algorithm on FF models, especially for the data owners who do not have full access to the models. Membership inference attacks (MIAs) Shokri et al. (2017) have been widely adopted as an empirical verification method for machine unlearning Gao et al. (2024), since other methods either sacrifice the model utility Sommer et al. (2022); Guo et al. (2023); Han et al. (2025) or necessitate full access Jagielski et al. (2022). However, current white-box MIAs are impractical for FF unlearning, as the data owners may not have full access to model parameters and gradients. Our experiments find that the existing black-box attacks are not accurate enough for FF models. Their effectiveness is often compromised by standard regularization techniques (e.g., dropout, batch normalization), which inherently decrease the attack success rate. This leads to the second key question in this paper: *How to design an accurate and practical verification method for FF unlearning algorithms?*

To address these challenges, we make the following contributions:

- *Problem Identification*: To the best of our knowledge, we are the first to formalize the problem and identify the unique challenges of machine unlearning for FF models. Direct gradient ascent induces optimization instability and frequent model collapse due to the sensitivity of FF models to parameter tuning. Layer-wise independent training further complicates the effectiveness-utility trade-off during unlearning.

- *Novel FF unlearning Framework*: We propose FF-Erase, the first unlearning framework specific to FF models. It introduces a novel goodness-guided approach where a dedicated guidance model directs layer-wise updates. We also propose two practical strategies to efficiently generate this guidance model, mini-retraining and fast-distillation, for a large amount and a small amount of remaining data, respectively.
- *Accurate Black-Box Unlearning Verification*: We propose a new black-box verification method for FF models, the goodness-based MIA (G-MIA). G-MIA leverages the unique properties of the FF models' goodness scores to achieve superior accuracy, providing a reliable tool for unlearning verification. We empirically demonstrate that G-MIA is effective when other black-box attacks fail with regulation techniques applied and even matches the performance of white-box attacks with deep networks and complex datasets.
- *Extensive Evaluation*: We demonstrate through extensive experiments that our method effectively unlearns target data while preserving model utility. FF-Erase achieves unlearning 1.9-3.1$\times$ faster than retraining from scratch, with only a minor 1.6-3.3% degradation in accuracy.

## 2 RELATED WORK

**Forward-Forward Algorithm:** The Forward-Forward algorithm (FF) Hinton (2022) was recently proposed as a novel training method to solve the bio-implausibility problem of backpropagation (BP) Rumelhart et al. (1986) methods, which are the dominant training methods for deep learning models. By eliminating the backward pass, FF models avoid storing intermediate activations and allow layers to process the next data batch immediately, thereby reducing memory consumption and enabling efficient pipeline parallelism. Therefore, numerous works have recently explored different FF algorithms. Initial efforts, such as Symba and Deeperforward Lee & Song (2023); Sun et al. (2025), focused on refining the core goodness function to support deeper networks and faster convergence. Building on these foundational improvements, subsequent work has expanded the FF training methods to more complex domains like convolutional (CwComp Papachristodoulou et al. (2024)), recurrent (FF-LSTM Gautham et al. (2024)), and graph-based (FORWARDGNN Park et al. (2024)) neural networks. As these FF algorithms investigate more complex tasks and architectures, the computational cost of retraining from scratch becomes increasingly prohibitive, creating an urgent need for efficient FF unlearning methods.

**Machine Unlearning:** Machine unlearning aims to remove the data impact of specific training samples from a trained model, while being efficient and preserving the utility of the unlearned model. Retraining the model from scratch is the gold standard for effectiveness and model utility, but it lacks efficiency. Existing works can be categorized into two types: exact and approximate unlearning. Exact unlearning methods seek to produce a model identical to the retrained model. However, current approaches are incompatible with general FF models, as they either rely on specific sharded architectures Bourtoule et al. (2021); Tao et al. (2024) or are restricted to linear models Guo et al. (2020). Approximate unlearning methods tune the model parameters to achieve fast forgetting. The dominant approaches perform gradient ascent (GA) on the forgetting data Tarun et al. (2023a); Sekhari et al. (2021a), while Qiao et al. (2024); Liu et al. (2022b); Wu et al. (2023b) refine this process by using techniques such as influence functions and Hessian matrix to estimate the parameter calibration. However, as discussed in §1 and Appendix §A, these methods were designed for BP-based models and are not suited for FF models due to their sensitivity to parameter tuning and risk of optimization instability. This leaves a clear gap for developing unlearning methods for FF models.

**Membership Inference Attacks:** Membership inference attacks (MIAs) Shokri et al. (2017); Nasr et al. (2019); Melis et al. (2019) are an empirical method for verifying the effectiveness of machine unlearning, particularly for complex, non-convex models Tu et al. (2024). The goal of an MIA is to determine if a given sample was in a model's training set. If an unlearning method is effective, MIAs should not successfully inference the forgetting samples as members. The more accurate an MIA is, the more reliable it is as a verification metric. MIAs are classified by their required level of access. White-box MIAs Wu et al. (2023a); Hamidouche et al. (2022) assume full access to model parameters and gradients, making them powerful but impractical for real-world verification, where data owners typically lack such privileged access or hardware resources for running full models. Black-box MIAs Liu et al. (2023); Cifuentes et al. (2021), which only use the model's final prediction output, are more practical but less accurate as a reliable verification metric. To fill this gap, we propose the Goodness-based MIA (G-MIA), a novel attack that leverages the unique layer-wise goodness scores of FF models. G-MIA achieves superior accuracy under a strict black-box constraint, being accurate and practical for verification.

## 3 PRELIMINARIES

In this section, we begin by reviewing the training and inference process of FF models in §3.1, and then formalize the machine unlearning problem and its notation in §3.2.

### 3.1 FORWARD-FORWARD TRAINING ALGORITHMS

**Data Forwarding and Goodness Calculation:** Consider a neural network model with $L$ layers for a $J$-class classification task. The objective of FF training is to optimize each layer $l$'s parameters $\theta^l$, so that every layer's goodness can better predict the correct class label $y$ for given input $\boldsymbol{x}$. Specifically, the function $f^l$ for each layer $l$ first computes its output $\boldsymbol{h}^l$ using its input $\boldsymbol{z}^{l-1}$ from layer $l-1$. Then it computes the goodness vector $\boldsymbol{g}^l$ based on $\boldsymbol{h}^l$, which reflects the activation degree of the neurons in a layer and is the key design for FF training and inference. After that, the layer simultaneously updates its parameters $\theta^l$ and forward $\boldsymbol{z}^l$, which is the normalization of $\boldsymbol{h}^{l1}$, to the next layer. Specially, the raw input $\boldsymbol{x}$ is considered as $\boldsymbol{z}^0$. This process is formalized as follows:

$$\boldsymbol{z}^0 = \boldsymbol{x}, \quad \forall l \in \{1, 2, \ldots, L\}, \quad \boldsymbol{h}^l = f^l(\boldsymbol{z}^{l-1}; \theta^l), \quad \boldsymbol{g}^l = \|\boldsymbol{h}^l\|_1, \quad \boldsymbol{z}^l = \frac{\boldsymbol{h}^l - \boldsymbol{g}^l}{\sqrt{\sigma^2 + \epsilon}}, \quad (1)$$

where $\sigma^2$ are the variance of $\boldsymbol{h}^l$ for layer normalization, and $\epsilon$ is a small constant to avoid dividing by zero. The goodness vector $\boldsymbol{g}^l = [g_1^l, g_2^l, \ldots, g_J^l]$ contains $J$ scores for each class, respectively.

**Loss Function and Optimization:** FF training aims to increase the goodness score $g_y^l$ of the correct class $y$ while suppressing the other goodness scores $g_{j,j\neq y}^l$. The loss function $\mathcal{L}_{\mathrm{ff}}$ is formalized as:

$$\forall l \in \{1, 2, \ldots, L\}, \quad \mathcal{L}_{\mathrm{ff}}(\boldsymbol{g}^l(\boldsymbol{x}, y; \theta^l)) = -\log\left(\frac{\exp\left(g_y^l\right)}{\sum_{j=1}^J \exp\left(g_j^l\right)}\right). \quad (2)$$

As this is a layer-wise loss function, FF training optimizes each layer's parameters independently:

$$\forall l, \quad \theta^l \leftarrow \theta^l - \eta \nabla_{\theta^l} \mathcal{L}_{\mathrm{ff}}(\boldsymbol{g}^l(\boldsymbol{x}, y; \theta^l)), \quad (3)$$

where $\eta$ is the learning rate. When optimizing $\mathcal{L}_{\mathrm{ff}}(\cdot)$, the distribution of layers' goodness vectors is shifting towards a direction where the goodness score of the correct class $g_y$ is significantly higher than others. For example, after training on data sample $(\boldsymbol{x}, y)$, the goodness distribution moves towards $\boldsymbol{g} = [g_1^{\sim}, g_2^{\sim}, \ldots, g_y^{\uparrow}, \ldots, g_J^{\sim}]$, where the uparrow $^\uparrow$ indicates significant increase and waves $\sim$ indicate moderate adjustments. As the average of goodness scores usually increases during FF training, we call this distribution shifting on goodness vectors as goodness increase for brevity.

**Model Inference:** FF models output the goodness vectors from all layers $\boldsymbol{g}^1, \boldsymbol{g}^2, \ldots, \boldsymbol{g}^L$ for inference. It is common to take a fully-connected layer on them as the predictor. We employ this predictor as our default setting in experiments due to its superior performance. We provide more details of the above FF training process using an illustration in Figure 2(a) for better understanding.

### 3.2 MACHINE UNLEARNING NOTATIONS

The purpose of a machine unlearning process is to remove the influence of forgetting data $\mathbb{D}_{\mathrm{forget}}$ from an original model $\theta_o$ (the model to unlearn) while maintaining the utility of unlearned model $\theta_u$ on the remaining data $\mathbb{D}_{\mathrm{remain}} = \mathbb{D}_{\mathrm{train}} \setminus \mathbb{D}_{\mathrm{forget}}$, where $\mathbb{D}_{\mathrm{train}}$ is the training dataset of $\theta_o$. Specifically, we denote the model retrained on $\mathbb{D}_{\mathrm{remain}}$ as $\theta_r$. This objective can be formalized as:

$$\min_{\theta^u \in \Theta} \mathcal{L}(\theta_u; \mathbb{D}_{\mathrm{forget}}) - \lambda \mathcal{L}(\theta_u; \mathbb{D}_{\mathrm{remain}}), \quad (4)$$

where $\lambda$ is a hyper-parameter to balance the trade-off between effectiveness, *i.e.*, loss value on forgetting data $\mathcal{L}(\theta_u; \mathbb{D}_{\mathrm{forget}})$ and model utility, *i.e.*, loss value on remaining data $\mathcal{L}(\theta_u; \mathbb{D}_{\mathrm{remain}})$.

---

[1] It is noted that $\boldsymbol{h}^l = [\boldsymbol{h}_1^l, \boldsymbol{h}_2^l, \ldots, \boldsymbol{h}_J^l]$ is a vector of vector, where each element $\boldsymbol{h}_1^l$ presents for the output vector of one class. The $\boldsymbol{h}^l$ is also denoted by the alias $\boldsymbol{H}^l \in \mathbb{R}^{J \times d^l J}$, where $d^l$ is the dimension of $l$-th layer's output. The $\boldsymbol{g}^l$ is calculated by the *column-wise L1 norm* of $\boldsymbol{h}^l$.

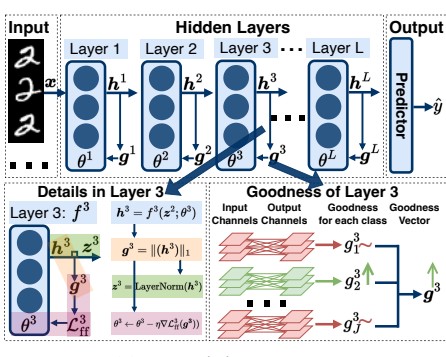

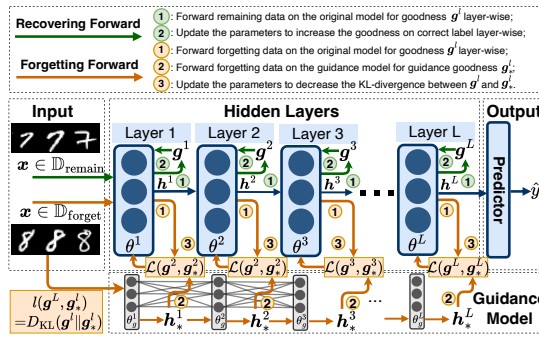

(a) FF training process     (b) FF-Erase unlearning process

Figure 2: Illustrations for FF learning (a) and FF-Erase unlearning (b). We elaborate the layer-wise training at the lower left corner and illustrate the multi-class goodness design at the lower right corner. For example, when training on images of number 2, the corresponding goodness score $g_2^3$ increases while others are suppressed. We also describe each step of unlearning at the upper corner.

## 4 METHODOLOGY

In this section, we first introduce the workflow of our proposed FF-Erase unlearning algorithm in §4.1. Then we present two practical strategies to efficiently acquire the guidance model required for performing FF-Erase unlearning in §4.2. Finally, we discuss the efficiency of FF-Erase in §4.3.

### 4.1 FAST FORWARD-FORWARD UNLEARNING

The key idea of FF-Erase unlearning is to decrease the goodness score on the forgetting data while maintaining the goodness score on the remaining data. The goodness decrease is the opposite process of learning, *i.e.*, $\boldsymbol{g} = [g_1^{\sim}, g_2^{\sim}, \ldots, g_y^{\downarrow}, \ldots, g_J^{\sim}]$ for forgetting data sample $(\boldsymbol{x}, y)$, which is named as "*forgetting forward*". To address the instability challenge during parameter tuning, we decrease the goodness under the guidance goodness $\boldsymbol{g}_*$ from a guidance model $\theta_g$, which is ignorant of the forgetting data but has the same architecture as the original model. Besides, we also run "*recovering forward*" to maintain the goodness score on the remaining data by repeating the learning process every $K$ epochs.[2] The overall workflow of FF-Erase unlearning is summarized as follows.

**Forgetting Forward**: 1) Every epoch, we forward the forgetting data samples through the original model and collect the goodness vector $\boldsymbol{g}_{(}\boldsymbol{x}; \theta)$; 2) we forward the same forgetting data samples through the guidance model to acquire the guidance goodness vector $\boldsymbol{g}_*(\boldsymbol{x}; \theta_g)$; 3) we decrease the goodness of forgetting data on the original model by minimizing the KL-loss between them:

---

**Algorithm 1** FF-Erase Unlearning Algorithm

**Input**: Models $\theta_o$ and $\theta_g$, epoch $E$, thresholds $\epsilon_1$ and $\epsilon_2$, datasets $\mathbb{D}_{\text{forget}}$ and $\mathbb{D}_{\text{remain}}$.
**Parameter**: FF model depth $L$, learning rate $\eta$, recovery step $K$, hyper-parameter $\lambda$.
**Output**: Unlearned model $\theta_u$.

1: **for** $e = 1, 2, \ldots, E$:
2:     **for** $\boldsymbol{x}$ in $\mathbb{D}_{\text{forget}}$:
3:         $\ell_1$=**FFwd**($\boldsymbol{x}, \theta_o, \theta_g$) // *forgetting forward*
4:     **for** $(\boldsymbol{x}, y)$ in $\mathbb{D}_{\text{remain}}$, **if** $e\%K == 0$:
5:         $\ell_2$=**RFwd**($\boldsymbol{x}, y, \theta_o$) // *recovering forward*
6:     **if** $\ell_1 < \epsilon_1$ **or** $\ell_2 > \epsilon_2$:   **break**

**Return**: $\theta_u = \theta_o$

**FFwd**($\boldsymbol{z}^0 = \boldsymbol{x}, \theta_o, \theta_g$):

1: **for** $l = 1, 2, \ldots, L$:
2:     $\boldsymbol{h}^l = f^l(\boldsymbol{z}^{l-1}; \theta_o^l)$, $\boldsymbol{h}_g^l = f^l(\boldsymbol{z}_g^{l-1}; \theta_g^l)$
3:     $\boldsymbol{z}^l = \textbf{LayerNorm}(\boldsymbol{h}^l)$, $\boldsymbol{z}_g^l = \textbf{LayerNorm}(\boldsymbol{h}_g^l)$
4:     $\boldsymbol{g}^l = \textbf{Norm}(\boldsymbol{h}^l)$, $\boldsymbol{g}_*^l = \textbf{Norm}(\boldsymbol{h}_g^l)$
5:     $\ell_1[l] = \nabla D_{\text{KL}}([\boldsymbol{g}^l], [\boldsymbol{g}_*^l])$, $\theta_o^l = \theta_o^l - \eta\ell_1[l]$
6: **return** $\sum_{l=1}^{L} \ell_1[l]$

**RFwd**($\boldsymbol{z}^0 = \boldsymbol{x}, y, \theta_o$):

1: **for** $l = 1, 2, \ldots, L$:
2:     $\boldsymbol{h}^l = f^l(\textbf{LayerNorm}(\boldsymbol{h}^{l-1}); \theta_o^l)$, $\boldsymbol{g}^l = \textbf{Norm}(\boldsymbol{h}^l)$
3:     $\ell_2[l] = \nabla\mathcal{L}_{\text{ff}}([\boldsymbol{g}^l], y)$, $\theta_o^l = \theta_o^l - \eta\lambda\ell_2[l]$
4: **return** $\sum_{l=1}^{L} \ell_2[l]$

---

$$\forall(\boldsymbol{x}, y) \in \mathbb{D}_{\text{forget}}, \forall l \in 1, 2, \ldots, L, \quad \theta^l \leftarrow \theta^l - \eta\nabla_{\theta^l} D_{\text{KL}}(\boldsymbol{g}^l(\boldsymbol{x}, y; \theta^l) \| \boldsymbol{g}_*^l(\boldsymbol{x}, y; \theta_g^l)), \quad (5)$$

---

[2]$K$ is an empirical hyper-parameter for model utility maintenance determined by the dataset. A smaller $K$ indicates more frequent recovering forwards, which usually leads to better model utility and worse efficiency.

which leverages a distillation-like manner for moderate parameter tuning during goodness decrease.

**Recovering Forward**: 1) Every $K$ epochs, we forward the remaining data samples through the original model and collect the goodness vector $\boldsymbol{g}(\boldsymbol{x}; \theta)$; 2) we update the parameters layer-wise to increase the goodness of remaining data. We summarize these two steps as:

$$\forall (\boldsymbol{x}, y) \in \mathbb{D}_{\text{remain}}, \forall l \in [1, L], \quad \theta^l \leftarrow \theta^l - \eta \nabla_{\theta^l} \lambda \mathcal{L}_{\text{ff}}(\boldsymbol{g}^l(\boldsymbol{x}, y; \theta^l)) \tag{6}$$

We provide more details to help understand the two forwards with corresponding steps including an illustration in Figure 2(b) and pseudocode in Algorithm 1. The functions **FFwd** and **RFwd** refer to the forgetting forward and recovering forward processes, respectively. We use **LayerNorm** and **Norm** to denote the layer normalization and $L_1$-norm operation for computing goodness in Equation (1), respectively. Rather than directly minimizing the goodness score of the correct class, FF-Erase decreases the goodness by shifting the goodness distribution towards the guidance goodness $\boldsymbol{g}_*$ using the Kullback-Leibler divergence for stable and moderate parameter tuning: $D_{\text{KL}}(\boldsymbol{g} \| \boldsymbol{g}_*) = \sum_{i=1}^{J} \hat{g}_i \log (\hat{g}_i / \hat{g}_{*i})$, where $\hat{g}_i = \exp g_i / \sum_{j=1}^{J} \exp g_j$ is the softmaxed goodness of the $i$-th class.

**Termination Conditions.** The unlearning process in FF-Erase will halt if the model fails to converge after a maximum number of epochs $E$. Besides, FF-Erase also employs an early stopping mechanism as commonly used in machine unlearning. Specifically, if the loss value update on the forgetting data $\mathbb{D}_{\text{forget}}$ drops below a threshold $\epsilon_1$ or the loss value on the remaining data $\mathbb{D}_{\text{remain}}$ exceeds a threshold $\epsilon_2$, FF-Erase will terminate unlearning and return the current model as the unlearned model $\theta_u$.

## 4.2 TRAINING GUIDANCE MODELS

To ensure both the efficiency and unlearning performance for the FF-Erase algorithm, we require a stable and accurate guidance model. That is to say, the guidance models need to provide stable guidance goodness distributions and be ignorant of the forgetting data. This is important for stabilizing the parameter calibration and avoiding model collapse during unlearning. Besides, the efficiency of generating the guidance model is also important. To this end, we propose two practical strategies to efficiently obtain accurate guidance models in different scenarios: mini-retrained and fast-distilled. Mini-retrained models are faster to obtain. However, when there are not enough remaining samples for retraining, we can still obtain fast-distilled models as slower alternatives, as they can be generated using fewer data samples.

**Mini-Retrained Strategy.** An ideal guidance model is one retrained from scratch on the remaining data, which is naturally stable and accurate. However, it is computationally prohibitive. As we do not demand guidance models' accuracy on the remaining data, we accelerate this process through two approximations: retraining $\alpha_1 = |\mathbb{D}_{\text{ref}}| / |\mathbb{D}_{\text{remain}}| \in (0, 1)$ proportion of the remaining samples using $\alpha_2 \in (0, 1)$ proportion of the epochs, where $\mathbb{D}_{\text{ref}} \subsetneq \mathbb{D}_{\text{remain}}$ is the selected subset:

$$\theta^{g,t} \leftarrow \theta^{g,t-1} - \eta \nabla_{\theta^{g,t-1}} \mathcal{L}(\mathbb{D}_{\text{ref}}; \theta^{g,t-1}). \tag{7}$$

**Fast-Distilled Strategy.** The knowledge distillation Hinton et al. (2015a); Gou et al. (2021) is a well-known approach to rapidly train a new model using existing models. Here, the original model $\theta_o$ acts as the "teacher". The goal is to train a "student" guidance model, $\theta_g$, to mimic the teacher's output on the remaining data. We use a simplified objective for fast distillation as follows:

$$\theta^{g,t} \leftarrow \theta^{g,t-1} - \eta \nabla_{\theta^{g,t-1}} D_{\text{KL}}(\mathbb{D}_{\text{ref}}; \theta^{g,t-1} \| \theta_o). \tag{8}$$

This strategy can also be accelerated using $\alpha_1$ and $\alpha_2$ as the mini-retrained strategy does.

## 4.3 EFFICIENCY OF FF-ERASE

The unlearning time of FF-Erase algorithm $t_{\text{unl}}$ contains two parts: the time to obtain the guidance model $t_0$ and the time for goodness decrease $t_1$. When unlearning $\beta = |\mathbb{D}_{\text{forget}}| / |\mathbb{D}_{\text{train}}| \in (0, 1)$ proportion of the training samples, the total time for FF-Erase using mini-retrained strategy is:

$$t_{\text{unl}} = t_0 + t_1 \approx \alpha_1 \cdot \alpha_2 \cdot t_{\text{ret}} + (K^{-1} + \beta) \cdot t_{\text{ret}}, \tag{9}$$

where $t_{\text{ret}}$ is the time for retraining from scratch. According to the experimental results in §6, we can achieve satisfactory unlearning performance using guidance models with $\alpha_1 = 0.3$ and $\alpha_2 = 0.5$, indicating an acceptable overhead of obtaining the guidance model (about 15% of $t_{\text{ret}}$). Empirically, $t_1$ usually takes another 10 to 20% of $t_{\text{ret}}$, leading to an overall $t_{\text{unl}}$ of 25 to 35% of $t_{\text{ret}}$ for FF-Erase to achieve effective unlearning. FF-Erase using fast-distilled strategy takes similar time.

## 5 GOODNESS-BASED MEMBERSHIP INFERENCE ATTACK (G-MIA)

In this section, we introduce the workflow of G-MIA and describe how to use G-MIA for quantitative verification of FF unlearning algorithms. We consider that the attacker can synthesize data that has a similar distribution to the training data, which is a common setting in related works (*e.g.*, Shokri et al. (2017); Liu et al. (2022a); Nasr et al. (2019)) and can be realized by model inversion techniques Fredrikson et al. (2015). It is also noted that the attacker can obtain the output of the target model of attack, *i.e.*, the goodness vectors from all layers. With the above information, a complete G-MIA contains the following four steps:

1) **Shadow Model Training.** The attackers first generate a synthetic dataset $\mathbb{D}_{\mathrm{syn}}$ and trains shadow models $\theta_{\mathrm{shadow}}$ on it. They also generate another separate synthetic dataset $\mathbb{D}'_{\mathrm{syn}}$ for testing.
2) **Goodness Feature Extraction.** The attacker collects the goodness vectors from all layers when member data ($\mathbb{D}_{\mathrm{syn}}$) and non-member data ($\mathbb{D}'_{\mathrm{syn}}$) of $\theta_{\mathrm{shadow}}$ forward the network.
3) **Attack Model Training.** The attacker uses the collected goodness vectors to train a binary classifier $f_{\mathrm{G-MIA}}(\cdot)$ that predicts whether a given sample is a member or non-member:

$$f_{\mathrm{G-MIA}}(\boldsymbol{g}^1, \boldsymbol{g}^2, \ldots, \boldsymbol{g}^L) = \begin{cases} 1, & member \\ 0, & non\text{-}member \end{cases} \tag{10}$$

4) **Membership Inference.** Given a specific data $\boldsymbol{d}$, the attacker first forwards $\boldsymbol{d}$ on the model under attack and obtains the goodness vectors, then predicts its membership by $f_{\mathrm{G-MIA}}(\cdot)$.

**G-MIA Verification**. We quantify the unlearning using the attack accuracy (ACC) and the area under the curve (AUC). A lower ACC or AUC score indicates fewer forgetting samples are identified as members, implying the unlearning is more effective. We provide more details in Appendix B.1.

## 6 EXPERIMENTS

In this section, we first present the effects of G-MIA in §6.1. Then we show the experimental results of FF-Erase unlearning regarding efficiency, effectiveness, and model utility in §6.2. In §6.3, we further explore classical unlearning methods under different parameters to robustly support our findings in §1. Lastly in §6.4, we present an ablation study to show the necessity and trade-offs of the guidance models. We evaluate FF unlearning on 4 standard image benchmarks: CIFAR-10, CIFAR-100 Krizhevsky et al. (2009), MNIST LeCun et al. (2010), and Fashion-MNIST Xiao et al. (2017), which are consistent with prior work on FF algorithms regarding the dataset complexity. We test on various FF models, including a 2-layer tiny CNN, AlexNet Krizhevsky et al. (2012), and VGG Simonyan et al. (2014) using state-of-the-art FF algorithms: CwComp and Deeperforward.

### 6.1 G-MIA PERFORMANCE

As an effective and reliable verification metric for FF unlearning, G-MIA should be accurate and present high ACC and AUC scores. To this end, we compare the attack accuracy (ACC) and area under the curve (AUC) of G-MIA with several state-of-the-art MIAs, including black-box final-layer MIA (FL) Shokri et al. (2017), white-box MIA using intermediate layer gradient (GR) Nasr et al. (2019), and white-box MIA using all layer outputs, including global average pooling (GAP) and statistics (ST). The statistics include mean, variance, maximum, and $L_2$ norm of all layer outputs. Our target models have employed basic MIA-defending techniques, including dropout, batch normalization, and weight decay. For each model, we randomly select 5000 pieces of data samples from the training set and test set, respectively, as the member and non-member data. The attack model for every type of MIAs is a standard multilayer perceptron with six hidden layers.

Our results shown in Figure 3 (using ACC as the metric[3]) indicate that G-MIA is an accurate and practical verification metric for FF unlearning. Firstly, G-MIA consistently outperforms the classical black-box final-layer MIA (FL) on all datasets and models. This indicates that the goodness from all layers provides more membership information than the final-layer output alone. Moreover, G-MIA even presents a better performance than white-box MIAs under deeper models and complex datasets. For example, G-MIA achieves the best accuracy under VGG13 and CIFAR-100. This is because deeper models and complex datasets amplify the impact of layer-wise independent training, making the goodness vectors from all layers more informative for membership inference.

---

[3]Due to space limitations, we show the results using AUC in the appendix §B.2.

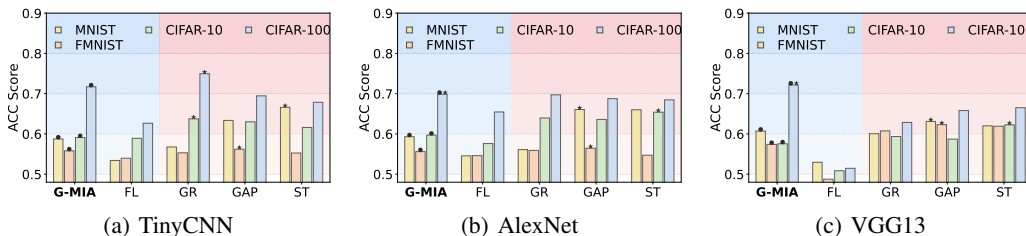

(a) TinyCNN        (b) AlexNet        (c) VGG13

Figure 3: Accuracy of different MIAs on various FF models. For each figure, we use a blue and red background to indicate black-box and white-box MIAs, respectively. We highlight the best black-box MIA using a circle and the best MIA of all types using a star.

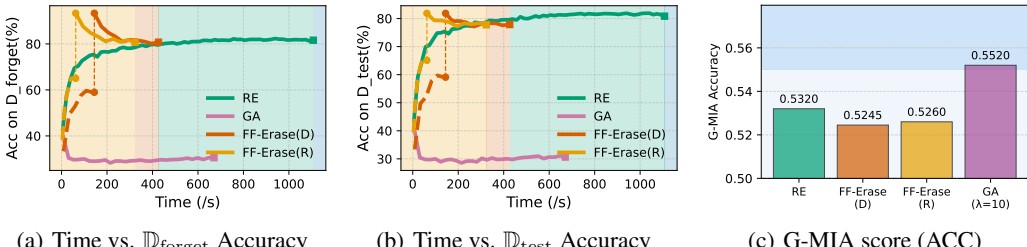

(a) Time vs. $\mathbb{D}_{\text{forget}}$ Accuracy     (b) Time vs. $\mathbb{D}_{\text{test}}$ Accuracy     (c) G-MIA score (ACC)

Figure 4: Unlearning performance comparison. In (a) and (b), the dashed lines of FF-Erase present the accuracy of guidance model. After generating the guidance model, the solid lines that follows is the accuracy of unlearned model. In (c), we quantify the effectiveness using G-MIA scores.

## 6.2 MACHINE UNLEARNING ON FF MODELS

In this experiment, we present the efficiency, effectiveness, and model utility of different unlearning methods using time versus accuracy curves. For the setup, we randomly sample 20% of the training data $\mathbb{D}_{\text{train}}$ as forgetting $\mathbb{D}_{\text{forget}}$ and use a separate test set, $\mathbb{D}_{\text{test}}$, from the same data distribution. As these three datasets share the same data distribution, effective unlearning algorithms will produce models that their accuracy on $\mathbb{D}_{\text{forget}}$ are similar to the original model's accuracy on $\mathbb{D}_{\text{test}}$. We further leverage G-MIA scores to rigorously quantify the extent of effective information removal. For model utility, a desirable algorithm should preserve performance of the unlearned model, meaning its accuracy on $\mathbb{D}_{\text{test}}$ should remain close to that of the original model.

We compare FF-Erase with retraining from scratch (RE) and direct gradient ascent (GA). RE is the gold standard for unlearning performance on effectiveness and model utility, while GA is a representative method for classical unlearning methods. We use FF-Erase(D) and FF-Erase(R) to denote FF-Erase unlearning using fast-distillation and retrain-based guidance models, respectively. Due to space limitations, we only show the results of VGG13 models trained on the CIFAR-10 dataset in the main text and put other results in Appendix §C.

As shown in Figure 4, our proposed FF-Erase efficiently realizes both effective and model utility. For effectiveness, FF-Erase(D) presents as a low G-MIA score (0.5245) as RE (0.532). It also achieves the same (81.31) accuracy on $\mathbb{D}_{\text{forget}}$ using only 38.52% of the RE time. For model utility, FF-Erase(D) retains similar accuracy as retaining on test data (80.85 and 77.87, respectively). Compared with FF-Erase(D), the FF-Erase(R) is more efficient (29.19% of the RE time) with tradeoffs on effectiveness (0.526 and 80.72 for G-MIA score and accuracy on $\mathbb{D}_{\text{forget}}$, respectively) and model utility (77.77). We also investigate the trade-off of different guidance models on FF-Erase in §6.4.

Noted that the GA method (when $\lambda = 10$) fails to converge and leads to model collapse. We will further explore its performance under different $\lambda$ choices in section §6.3.

## 6.3 FURTHER EXPLORATION ON CLASSICAL UNLEARNING METHODS

In this experiment, we further explore the impact of different $\lambda = 10^1, 10^0, 10^{-1}, 10^{-2}, 10^{-3}, 0$ in Equation (4) on the performance of classical unlearning methods using gradient ascent as a representative. Our results on VGG13 models trained on the CIFAR-10 dataset are shown in Figure 5, indicating that the model will either collapse ($\lambda = 10^1, 10^0, 10^{-1}$) or cannot unlearn the forgetting

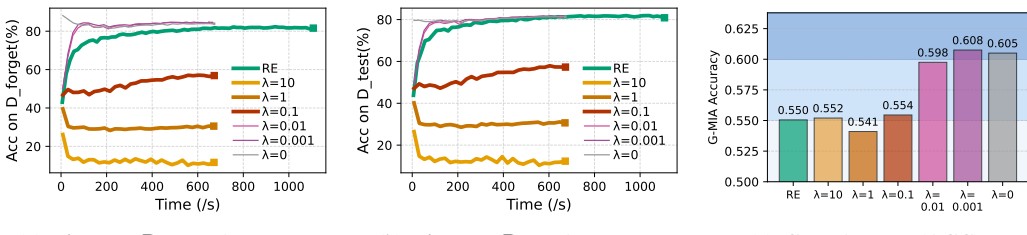

(a) Time vs. $\mathbb{D}_{\text{forget}}$ Accuracy     (b) Time vs. $\mathbb{D}_{\text{test}}$ Accuracy     (c) G-MIA score (ACC)

Figure 5: GA performance under different $\lambda$. Some GA methods($\lambda = 0.01, 0.001, 0$) are ineffective.

data ($\lambda = 10^{-2}, 10^{-3}, 0$). As shown in Figure 5(a), GA (when $\lambda = 10^{-2}, 10^{-3}, 0$) presents significantly high accuracy on forgetting data $\mathbb{D}_{\text{forget}}$ (84.43, 83.3, and 83.32, respectively) compared with RE (81.61) and FF-Erase(D) (81.31 in Figure 4(a)), indicating poor unlearning effectiveness. Figure 5(c) shows a more precise result, where GA ($\lambda = 10^{-2}, 10^{-3}, 0$) gets G-MIA scores of 0.6, 0.61, and 0.6, respectively, being much higher than RE (0.55). For model utility in Figure 5(b), GA (when $\lambda = 10^1, 10^0, 10^{-1}$) shows low accuracy $\mathbb{D}_{\text{test}}$ (below 60), failing to preserve model utility.

## 6.4 ABLATION STUDY ON GUIDANCE MODELS

In this experiment, we explore the efficiency-performance trade-off of different guidance models in FF-Erase. While a faster acquisition of guidance model can accelerate the unlearning process, it may sacrifice the unlearning effectiveness and model utility. We test different proportions of selected data $\alpha_1$ and of epochs $\alpha_2$ for acquiring guidance models and utilize them for FF-Erase unlearning.

| Methods | Efficiency | | | Effectiveness | | | Utility |
|---|---|---|---|---|---|---|---|
| | $t_{\text{unl}}$ (/s)↓ | $t_0$ (/s)↓ | $t_{\text{unl}} - t_0$ (/s)↓ | $\text{Acc}_{\text{f}}$ (%)↓ | G-MIA ACC↓ | G-MIA AUC↓ | $\text{Acc}_{\text{t}}$ (%)↑ |
| RE | 1107 | 0 | 1107 | 81.61 | 0.551 | 0.571 | 80.85 |
| D-(0.5,0.5) | 583.5 | 410.5 | 173 | 81.58 | 0.556 | 0.577 | 78.34 |
| D-(0.5,0.2) | 426.7 | 158.9 | 267.8 | 80.76 | 0.561 | 0.583 | 77.95 |
| D-(0.5,0.1) | 353.7 | 83.7 | 270 | 80.48 | 0.587 | 0.612 | 77.54 |
| D-(0.3,0.5) | 569.6 | 288.5 | 281.1 | 81.12 | 0.568 | 0.595 | 77.87 |
| D-(0.3,0.2) | 391.1 | 111.7 | 279.4 | 80.73 | 0.571 | 0.606 | 77.81 |
| R-(0.5,0.5) | 518.5 | 155.1 | 363.4 | 81.53 | 0.562 | 0.583 | 79.16 |
| R-(0.5,0.2) | 443.7 | 60.7 | 383 | 81.02 | 0.573 | 0.609 | 79.00 |
| R-(0.3,0.5) | 474.4 | 107.8 | 366.6 | 81.51 | 0.569 | 0.598 | 78.86 |
| R-(0.3,0.2) | 429.6 | 41.8 | 387.6 | 81.03 | 0.577 | 0.621 | 78.58 |
| R.G.M | 950.8 | 0 | 950.8 | 51.18 | 0.553 | 0.575 | 55.53 |

Table 1: FF-Erase unlearning using different guidance models. $\text{Acc}_{\text{f}}$ and $\text{Acc}_{\text{t}}$ respectively denote the accuracy on $\mathbb{D}_{\text{forget}}$ and $\mathbb{D}_{\text{test}}$. D (R) refers to fast-distilled (mini-retrained) strategy, followed by $\alpha_1$ and $\alpha_2$, e.g., D-(0.5,0.1) refers to FF-Erase guided by a fast-distilled guidance model on $\alpha_1$=50% data for $\alpha_2$=10% epochs. The $t_{\text{unl}}$ is the total unlearning time containing two parts: guidance model obtaining $t_0$ (if any) and goodness decrease $t_{\text{unl}} - t_0$. R.G.M in the last line refers to FF-Erase using randomly initialized guidance model. The ↓ (↑) indicates that a lower (higher) is better.

Firstly, as shown in Table 1, a stable and accurate guidance model is crucial for FF-Erase unlearning. FF-Erase using a randomly initialized model as guidance model (denoted as R.G.M in the last line) leads to unacceptable performance degradation: the $\text{ACC}_{\text{t}}$ drops to 55.53%. Such a guidance model could not provide stable guidance goodness for goodness decreasing, leading to a situation similar to the direct gradient ascent (GA) method. Secondly, using more data samples for generating the guidance model (a larger $\alpha_1$) leads to better unlearning performance (lower G-MIA ACC, i.e., more effective, and higher $\text{ACC}_{\text{t}}$, i.e., better model utility) but requires longer unlearning time $t_{\text{unl}}$. Using more training epochs (a larger $\alpha_2$) also leads to the same trend. Our ablation study demonstrates that FF-Erase can flexibly achieve different efficiency-performance trade-offs by choosing different guidance strategies and hyperparameters, making it adaptable to various application scenarios.

## 7 CONCLUSION

In this paper, we propose FF-Erase, the first machine unlearning method for FF models. We identify the problem that existing unlearning methods designed for BP-based models are infeasible for FF models due to the sensitivity of FF models to parameter changes. To address this challenge, we design FF-Erase, a novel FF-specific gradient ascent method to effectively erase the data impact of

forgetting samples. FF-Erase uses a goodness-based regularization to stabilize the parameter calibration and a layer-wise unlearning scheme to promote the unlearning efficiency. Moreover, we propose two flexible strategies to acquire the guidance model for FF-Erase. Accordingly, we propose G-MIA, a goodness-based membership inference attack, to quantitatively verify the unlearning effectiveness of FF-Erase. Extensive experiments on various datasets and model architectures demonstrate that FF-Erase is effective and efficient, achieving comparable unlearning effectiveness as retraining while being 1.9-3.1$\times$ faster.

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

## A    SUPPLEMENTARY ON RELATED WORK

In this appendix section, we specifically discuss more details of related works on conventional approximate unlearning methods for BP models and explain why they are not suitable for FF models as mentioned in §1 and §2. Moreover, in the main text, we utilize direct gradient ascent (GA) as the representative of the approximate unlearning methods. Here in this appendix section, we also explain the rationale of the GA's representativeness.

**Robust Gradient Ascent:** While directly applying GA by increasing the loss on forgetting data can be too brute-force and damages the model's utility, advanced GA methods use different techniques for realizing a more robust GA, and those methods are proven to be successful in BP models. Tarun et al. (2023b) introduces an impair-and-repair process, adding gradient descent on remaining data after all GA epochs to repair the model utility. Gao et al. (2024) balances the utility using a scale-to-unlearn technique. Sepahvand et al. (2025) leverages adversarial training to unlearn representations of forgetting data. However, all those methods face the same failure for FF models as direct GA: the model collapse takes place immediately, making those repairing techniques ineffective, as experimental results in Appendix §C.3 have demonstrated. This is because all those methods only limit how much the gradient ascent is; however, the FF models additionally require avoiding invalid goodness distribution, which is not considered in those methods. Therefore, those robust GA methods still can not achieve effective FF unlearning without model collapse.

**Distillation-based Gradient Ascent:** The teacher-student approaches, or distillation-based GA methods Kurmanji et al. (2023); Chundawat et al. (2023a); Bagheri et al. (2025); Wang et al. (2025a); Wu et al. (2022), are another type of advanced GA methods leveraging knowledge distillation Hinton et al. (2015b) techniques for a gentle gradient ascent. However, these methods are also not feasible to be directly applied to FF models for two specific reasons. First, they depend on the final-layer logits for unlearning, which is not effective enough, as FF models also persist knowledge in their previous layers during their layer-wise greedy learning. Second, literature such as SCRUB Kurmanji et al. (2023) utilizes the original model as the teacher model and unlearns by increasing the divergence between the original and unlearned model outputs on forgetting data. Although this achieves successful unlearning for BP models, it can lead to an away-but-invalid goodness distribution for FF models, risking model collapse. Therefore, those methods still can not achieve effective FF unlearning without model collapse.

## B    GOODNESS-BASED MEMBERSHIP INFERENCE ATTACK (G-MIA)

In this appendix section, we will first provide supplementary descriptions of G-MIA verification as mentioned in §5. Then we will show more experimental results of G-MIA to demonstrate its accuracy and lightweight as mentioned in §6.1. Finally, we provide the G-MIA score (AUC) of different unlearning methods under various settings as mentioned in §6.2 and §6.3.

### B.1    G-MIA VERIFICATION

Besides attack accuracy (ACC), the area under the receiver operating characteristic curve (AUC) is also widely used in related work Shokri et al. (2017); Carlini et al. (2022); Liu et al. (2022c); Shi et al. (2024) to quantify the accuracy of an MIA. AUC describes the probability that a randomly chosen member receives a higher score than a randomly chosen non-member. Let $f_{\mathrm{MIA}}(\boldsymbol{x}) \in [0, 1]$ be a continuous MIA prediction score, *i.e.*, the attack model outputs a float number from 0 to 1 to measure the likelihood that $\boldsymbol{x}$ is a member, rather than directly 0 as a non-member and 1 as a member. The AUC can be calculated by:

$$\mathrm{AUC} = P[f_{\mathrm{MIA}}(X_M) > f_{\mathrm{MIA}}(X_N)] + 0.5 \times P[f_{\mathrm{MIA}}(X_M) = f_{\mathrm{MIA}}(X_N)], \quad (11)$$

where $X_M$ and $X_N$ denote member and non-member data, respectively. The result from the above definition is equivalent to the area under the receiver operating characteristic (ROC) curve Fawcett (2006), where ROC plots True Positive Rate (TP) vs. False Positive Rate (FP) as the threshold varies across all real values. For verification, a lower ACC or AUC score on the forgetting data indicates that these data are less likely to be detected as members, thereby demonstrating a more effective unlearning method. Existing literature has rigorously analyzed the relationship between MIA ACC

(AUC) scores and unlearning effectiveness Tu et al. (2024), which is orthogonal and can be applied to our work.

In our experiments, we use both of them to quantify the unlearning. We utilize ACC because it is more intuitive and easier to understand. However, it is sensitive to the member/non-member ratio in the evaluation and requires a specific threshold to classify the membership. Therefore, we also leverage the AUC score, which is invariant to the ratio, thereby being easier to compare across datasets, splits, or output interfaces. Due to the space limitation, we only give the ACC results in the main text. Then we will show the corresponding AUC results in §B.2.

### B.2 G-MIA PERFORMANCE

Firstly, we give the AUC score of different MIAs on various FF models in Figure 6, which corresponds to the ACC results in Figure 3. The AUC results further demonstrate that G-MIA is an accurate and practical verification metric for FF unlearning. G-MIA consistently outperforms the classical black-box final-layer MIA (FL) on all datasets and models. Moreover, G-MIA even presents a better performance than white-box MIAs under deeper models and complex datasets. For example, G-MIA achieves the best accuracy under VGG13 and CIFAR-100.

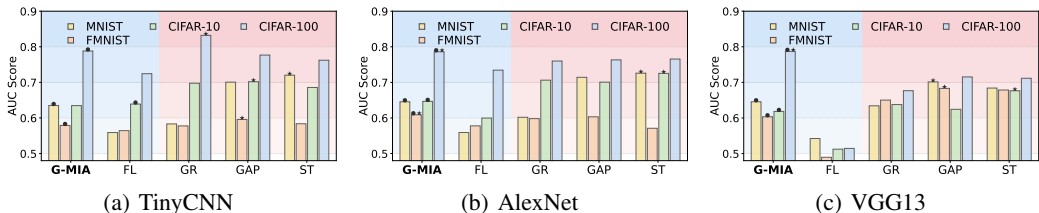

(a) TinyCNN  (b) AlexNet  (c) VGG13

Figure 6: AUC score of different MIAs on various FF models in Figure 3. For each figure, we use a blue and red background to indicate black-box and white-box MIAs, respectively. We highlight the best black-box MIA using a circle and the best MIA of all types using a star.

Besides, we also give a detailed comparison of the input data size required by different types of MIAs in Table 2, which demonstrates that G-MIA is a lightweight attack. FL, G-MIA, and GD only require small amounts of input data ($10^1 \sim 10^3$), while GAP and ST need much larger input data ($10^3 \sim 10^5$). The input data size directly determines the size of the attack model, including training efficiency and memory consumption. Therefore, G-MIA is not only accurate but also lightweight.

| Setting | FL | G-MIA | GAP | ST | GR |
|---|---|---|---|---|---|
| C10/MN/FMN-VGG16 | 10 | 170 | 12880 | 38657 | 103 |
| CIFAR100-VGG16 | 100 | 1700 | 18400 | 55217 | 103 |
| C10/MN/FMN-VGG13 | 10 | 140 | 4760 | 14294 | 85 |
| CIFAR100-VGG13 | 100 | 1400 | 6800 | 20414 | 85 |
| C10/MN/FMN-AlexNet | 10 | 50 | 1300 | 3905 | 31 |
| CIFAR100-AlexNet | 100 | 500 | 1300 | 3905 | 31 |
| C10/MN/FMN-TinyCNN | 10 | 40 | 1100 | 3304 | 25 |
| CIFAR100-TinyCNN | 100 | 400 | 1100 | 3304 | 25 |

Table 2: Input data size required by different types of MIAs. We use C10/MN/FMN to refer to CIFAR10, MNIST, or FMNIST for brevity.

### B.3 G-MIA AUC SCORE IN EXPERIMENTS

In §6.2 and §6.3, we have shown the G-MIA ACC score of different unlearning methods under various settings in Figure 4(c) and 5(c), respectively. Here we provide the corresponding AUC score in Figure 7(a) and (b), respectively. The AUC results are consistent with the ACC results, which further demonstrate that our proposed method is effective for FF unlearning, and demonstrate that GA methods (when $\lambda = 0.01, 0.001, 0$) can not effectively unlearn the forgetting data.

## C SUPPLEMENTARY EXPERIMENTAL RESULTS

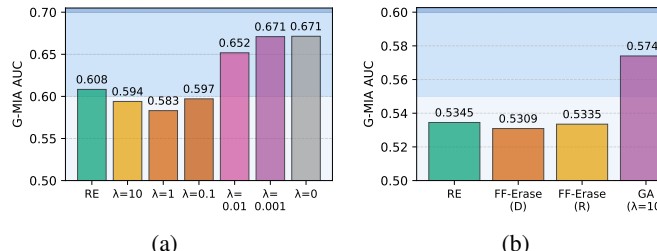

(a)           (b)

Figure 7: G-MIA score (AUC). Figure (a) and (b) provide the GIA AUC score of Figure 4(c) and 5(c), respectively. The AUC score of the original model (0.6184) in Figure (b) provides a reference, that any unlearning method with a lower G-MIA AUC score is regarded as effective.
.

In this appendix section, we provide more experimental results. Firstly, in §C.1, we extend the results of different datasets and FF models to demonstrate the generalizability of our findings in §6. Then in §C.2, we show the results of a new experiment, which utilizes centered kernel alignment (CKA) similarity to evaluate the effectiveness layer-wise, indicating the effectiveness of our FF-Erase. After that, in §C.3, we provide extra experiments using more unlearning methods as baselines to further validate the superiority of our proposed FF-Erase. Finally, in §C.4, we explore the training curve of guidance models to explore their characteristics.

## C.1 RESULTS UNDER DIFFERENT SETTINGS FOR EXPERIMENTS IN MAIN PAPER

While the main paper presented results for VGG13 on CIFAR-10 due to space constraints, this appendix provides supplementary experiments across different datasets and various FF models to demonstrate the generalizability of our findings.

Figure 8 extends our evaluation to CIFAR-100, MNIST, and Fashion-MNIST. These results corroborate the main paper's findings: FF-Erase is significantly more efficient than RE while achieving comparable unlearning effectiveness and model utility. Moreover, GA-based methods consistently fail to converge on these datasets. We observe that the required unlearning epochs (the hyperparameter $\alpha_2$) correlates with dataset complexity. Compared with the more complex CIFAR-100, both FF-Erase(D) and FF-Erase(R) require fewer unlearning epochs on simpler datasets like MNIST and Fashion-MNIST. This is expected, as models trained on complex data learn more data features, making the information removal of forgetting data more difficult.

Similarly, Figure 9 presents results across different FF model architectures (TinyCNN, AlexNet, and VGG16). These experiments confirm that the conclusions regarding the efficiency, effectiveness, and utility of our proposed method hold irrespective of the underlying model architecture.

## C.2 LAYER-WISE UNLEARNING

As FF models are trained using a layer-wise and greedy optimization, different layers may retain different aspects of knowledge from the training data. To empirically show the effectiveness of our proposed FF-Erase in removing residual knowledge from all layers, we further evaluate the layer-wise unlearning performance using Centered Kernel Alignment (CKA) Wang et al. (2015); Kornblith et al. (2019) similarity as the metric.

CKA is a similarity measure between two sets of representations, which has been widely used in analyzing the unlearning effectiveness of different layers in neural networks Xu et al. (2025); Kim et al. (2025). Given two representation matrices $X_i^o \in \mathbb{R}^{n \times p}$ from original model and $X_i^u \in \mathbb{R}^{n \times p}$ from unlearned model, where $n$ is the number of samples, and $p$ are the dimensions of the output representations of layer $i$, the linear CKA similarity between them is calculated by:

$$\text{CKA}(X_i^o, X_i^u) = \frac{\text{HSIC}(X_i^o, X_i^u)}{\sqrt{\text{HSIC}(X_i^o, X_i^o)\text{HSIC}(X_i^u, X_i^u)}}, \tag{12}$$

where $\text{HSIC}(\cdot, \cdot)$ is the Hilbert-Schmidt Independence Criterion. Detailed calculations of HSIC can be found in Wang et al. (2015). The CKA similarity ranges from 0 to 1, where a lower value indicates

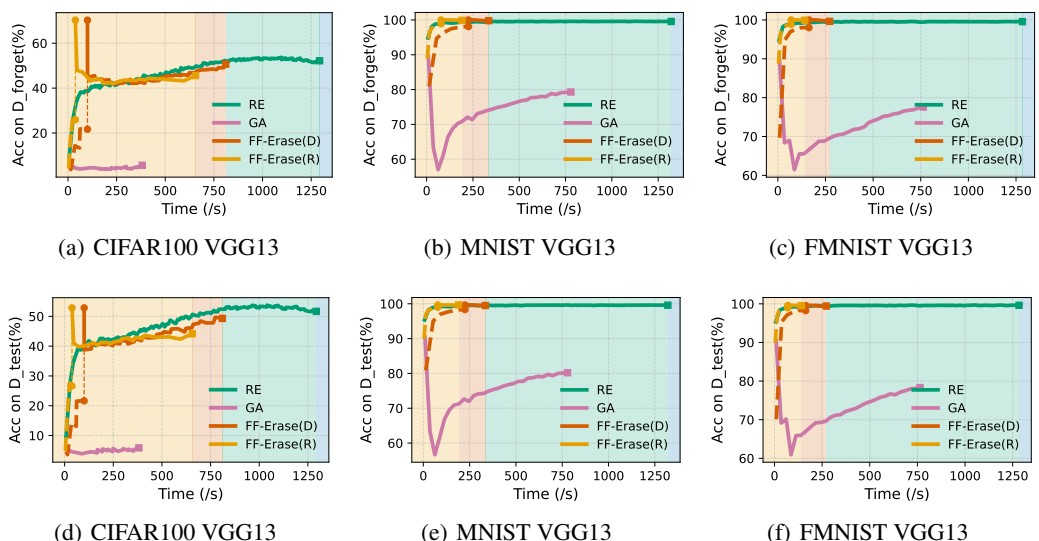

Figure 8: Comparison among different FF unlearning methods under different datasets. Figures in the first line (a), (b), and (c) show the model accuracy on forgetting data $\mathbb{D}_{\text{forget}}$ and the figures in the second line (d), (e), and (f) show the model accuracy on test data $\mathbb{D}_{\text{test}}$.

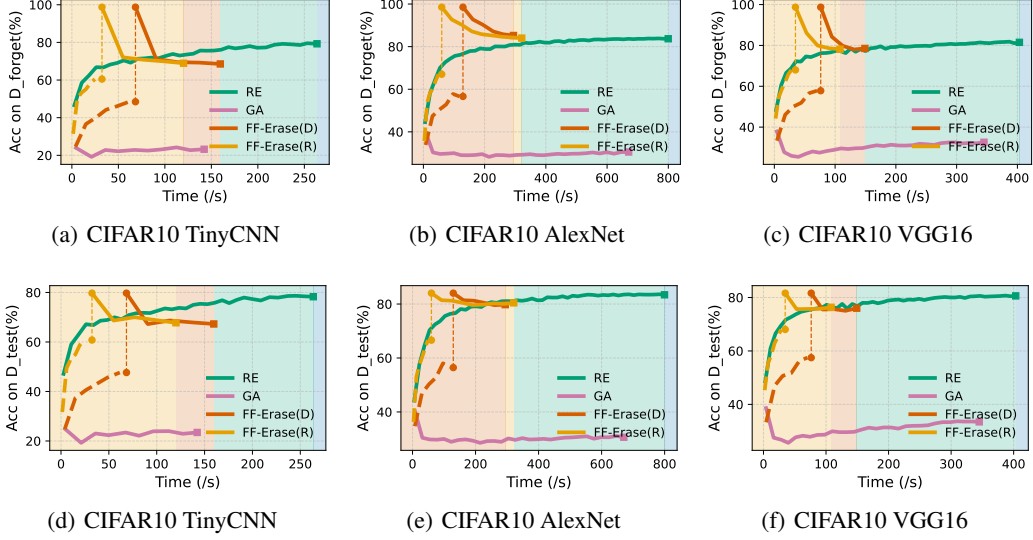

Figure 9: Comparison among different FF unlearning methods under different FF models. Figures in the first line (a), (b), and (c) show the model accuracy on forgetting data $\mathbb{D}_{\text{forget}}$ and the figures in the second line (d), (e), and (f) show the model accuracy on test data $\mathbb{D}_{\text{test}}$.

a larger difference between the two representations. We compare the CKA similarity on forgetting data between the original and unlearned models, including FF-Erase(D), FF-Erase(E), retraining (RE), and direct gradient ascent (GA), following the same setting of Figure 4. We additionally add extra baselines: Bad Teacher (BT) Chundawat et al. (2023a), an advanced distillation-based gradient ascent method of approximate unlearning; FATS Tao et al. (2024), an advanced retraining-based exact unlearning; FYE Tarun et al. (2023b), a robust gradient ascent method. The CKA scores of RE show an unlearning standard. The results are shown in Table 3 as follows: Firstly, for the retraining model (RE), the CKA similarity scores first decrease and then increase along the layer depth. For shallow layers, CKA scores are highest, indicating that those layers learn general features shared by both forgetting and remaining data, making them hard to unlearn even by retraining. For middle layers, CKA scores are lowest, implying that those layers learn more specific features of forgetting

| Layer | 1 | 2 | 3 | 4 | 5 | 6 | 7 |
|---|---|---|---|---|---|---|---|
| RE | 0.9918 | 0.9249 | 0.8004 | 0.6319 | 0.6286 | 0.5700 | 0.3619 |
| FF-Erase(D) | 0.9958 | 0.9658 | 0.9066 | 0.8100 | 0.7584 | 0.7309 | 0.6168 ↑ |
| FF-Erase(R) | 0.9961 | 0.9721 | 0.9199 | 0.8279 | 0.7816 | 0.7454 | 0.6292 ↑ |
| GA($\lambda$=10) | 0.5809 ↓ | 0.3886 ↓ | 0.2915 ↓ | 0.2454 ↓ | 0.2308 ↓ | 0.1892 ↓ | 0.1476 ↓ |
| GA($\lambda$=$10^{-2}$) | 0.9952 | 0.9704 | 0.9219 | 0.8368 ↑ | 0.7801 | 0.7404 | 0.6152 ↑ |
| BT | 0.9998 | 0.9975 | 0.9915 | 0.9638 ↑ | 0.9572 ↑ | 0.9008 ↑ | 0.7571 ↑ |
| FYE | 0.9827 | 0.6004 ↓ | 0.2147 ↓ | 0.1242 ↓ | 0.1003 ↓ | 0.0571 ↓ | 0.0331 ↓ |
| FATS | 0.9723 | 0.9118 | 0.7971 | 0.6772 | 0.6478 | 0.5677 | 0.3827 |
| Layer | 8 | 9 | 10 | 11 | 12 | 13 | - |
| RE | 0.3472 | 0.4981 | 0.4200 | 0.4923 | 0.6223 | 0.6144 | - |
| FF-Erase(D) | 0.5737 ↑ | 0.5462 | 0.4548 | 0.5071 | 0.5742 | 0.5488 | - |
| FF-Erase(R) | 0.5919 ↑ | 0.5930 | 0.5128 | 0.5748 | 0.6490 | 0.6358 | - |
| GA($\lambda$=10) | 0.1317 ↓ | 0.1495 ↓ | 0.1255 ↓ | 0.1116 ↓ | 0.0867 ↓ | 0.0656 ↓ | - |
| GA($\lambda$=$10^{-2}$) | 0.5562 ↑ | 0.5888 | 0.4809 | 0.5568 | 0.6309 | 0.6144 | - |
| BT | 0.6368 ↑ | 0.7053 ↑ | 0.5766 | 0.6097 | 0.7573 | 0.0049 ↓ | - |
| FYE | 0.0295 ↓ | 0.0262 ↓ | 0.0150 ↓ | 0.0075 ↓ | 0.0072 ↓ | 0.0056 ↓ | - |
| FATS | 0.3312 | 0.4502 | 0.3564 | 0.3713 | 0.5776 | 0.5944 | - |

Table 3: Layer-wise CKA similarity between original and unlearned models on forgetting data under the VGG13 and CIFAR-10 settings of Figure 4. We use ↑ (↓) to denote CKA scores that are 20% higher (lower) than the RE scores, which suggests ineffective unlearning (over forgetting).

data, which can be effectively unlearned by retraining. It might also suggest unique characteristics of FF models, where middle layers are more specialized for each training process. For deep layers, CKA scores slightly increase again, which may be because deep layers learn high-level features shared by both forgetting and remaining data, as our forgetting data are randomly sampled rather than unlearning specific classes of data.

Secondly, for FF-Erase methods, the CKA scores show a clear trend of decreasing similarity across all layers, indicating their effectiveness in unlearning the forgetting data. This is particularly evident in FF-Erase(D), which consistently outperforms FF-Erase(R) in all layers. The results suggest that FF-Erase(D) is more effective in erasing the specific features learned from the forgetting data, while FF-Erase(R) retains some of those features, leading to higher CKA scores.

Thirdly, for conventional approximate unlearning methods, the layer-wise CKA scores are consistent with our conclusion: some lead to model collapse (*e.g.*, GA($\lambda = 10$) and FYE) and some lead to ineffective unlearning (*e.g.*, GA($\lambda = 10^{-2}$) and BT). For GA, a direct gradient ascent method relying on loss on forgetting data, the CKA scores (except the first layer) decrease sharply, which is reasonable since a collapsed model (model unlearned by GA) is not supposed to extract similar representations as a well-trained model (the original model). For BT, a distillation-based gradient ascent method, its CKA scores show why it is ineffective for FF models in two aspects: the shallow and middle layers retain too much information on forgetting data (a significantly higher score compared to RE for each layer), while the last layer extracts totally different representations. Such a model state, where residual knowledge persists in shallow/middle layers while over-forgetting happens in deep layers, will be easily captured by G-MIA attacks.

Lastly, for FATS, a retraining-based exact unlearning method, its CKA scores are close to RE, indicating its effectiveness. However, as shown in §C.3, FATS still requires a large amount of unlearning time, making it inefficient compared to our proposed FF-Erase.

## C.3 Supplementary Experiments

In §6, we use direct gradient ascent (GA) as the representative of conventional BP unlearning methods. Here we provide supplementary experimental results covering additional baselines, including robust GA methods FYE Tarun et al. (2023b) and SURE Sepahvand et al. (2025), distillation-based GA method Bad Teacher (BT) Chundawat et al. (2023a), and advanced retraining-based method FATS Tao et al. (2024). FATS utilizes incremental learning to bypass unnecessary retraining epochs; its expectation of unlearning time for a single data point is half of RE. It is noted that those new baselines often trigger our termination conditions in §4.1, including timeout (exceeds a maximum

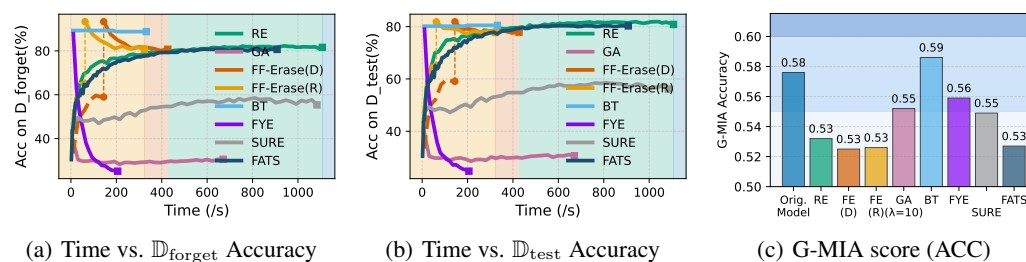

(a) Time vs. $\mathbb{D}_{\text{forget}}$ Accuracy     (b) Time vs. $\mathbb{D}_{\text{test}}$ Accuracy     (c) G-MIA score (ACC)

Figure 10: Extension of Figure 4, adding extra baselines. The dashed lines are present for generating guidance models, which are required by FF-Erase methods. The solid lines represent the accuracy of the unlearned model. Other baselines do not need a guidance model, so they only have solid lines. In (c), we shortened some baseline names for the second time due to the space limit, including "Orig. Model" for "Original Model" and "FE" for "FF-Erase".

number of epochs E determined by RE time), loss plateau (no loss decrease on remaining data for 10 consecutive epochs), and utility collapse (accuracy on remaining data lower than a threshold 25%). The results are consistent with our description in shown in §1 and §2, showing in Figure 10 as follows:

Firstly, since we are unlearning multiple data points (20% of the training set), the advanced retraining method FATS still requires a large amount of unlearning time (around 900s), which is close to RE (around 1100s). Therefore, it is not as efficient as our proposed FF-Erase.

Secondly, for robust GA methods, FYE (terminated at around 200s by utility collapse) and SURE (terminated at around 1100s by timeout) still lead to model collapse as direct GA does. This experimental result is consistent with our explanation in §4.1, that FF unlearning requires not only decreasing goodness scores on forgetting data but also maintaining a valid goodness distribution. Although those robust GA methods achieve great success in BP unlearning, they are not feasible for FF unlearning without considering the unique characteristics of FF models.

Thirdly, the distillation-based GA method, Bad Teacher (BT)(terminated at around 330s by loss plateau), still leads to ineffective unlearning as direct GA does (when $\lambda$ is small). The G-MIA accuracy on BT method (0.59) is significantly higher than others ($\leq$ 0.58), demonstrating its ineffectiveness in quantitative. This experimental result is consistent with our explanation in §A, that those distillation-based GA methods are not suitable for FF unlearning due to their dependence on final-layer logits.

Overall, these supplementary experiments confirm the representativeness of direct GA as the baseline of conventional BP unlearning methods in §6. Comparisons with those additional baselines further validate our analysis in §1 and §2: exact unlearning methods are too time-consuming, while approximate unlearning methods (represented by direct gradient ascent, GA) lead to either model collapse or ineffective unlearning.

## C.4 DETAILS OF GUIDANCE MODEL

In this section, we explore the training curve of guidance model training used in §6.4. All the guidance models have a much lower accuracy on various datasets regardless the training time, and thus being not suitable to directly serve as the unlearned model. However, when we apply our strategies in §4.2, we can efficiently obtain guidance models for FF-Erase unlearning. The details are as follows:

Firstly, even given sufficient training time, all mini-retrained guidance models ($\leq$75%) and fast-distilled guidance models ($\leq$70%) have a significantly lower accuracy on test set compared to the retraining methods RE (81%). The guidance models used in FF-Erase have a worse accuracy (60%-71%) than corresponding sufficiently-trained cases, because they are picked before converging to reduce the overall unlearning time.

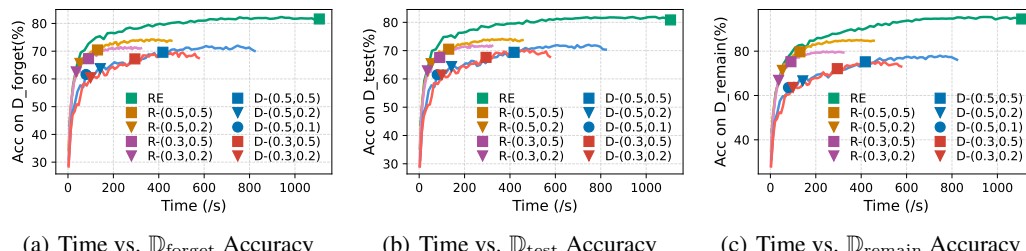

(a) Time vs. $\mathbb{D}_{\mathrm{forget}}$ Accuracy     (b) Time vs. $\mathbb{D}_{\mathrm{test}}$ Accuracy     (c) Time vs. $\mathbb{D}_{\mathrm{remain}}$ Accuracy

Figure 11: Training curve of guidance models used in Table 1. The curve shows the accuracy of the models on different datasets over time. Figures (a), (b), and (c) shows the accuracy on forgetting data, test data and remaining data, respectively. The five lines respectively presents retraining, mini-retraining using 0.5 of remaining data, mini-retraining using 0.3 of remaining data, fast-distilling using 0.5 of remaining data, and fast-distilling using 0.3 of remaining data. The points on the curve indicates the guidance model we pick up. For example, the blue triangle represents taking 20% of the training epochs for fast-distilling using 0.5 of the remaining data.

Secondly, the guidance models are efficient to obtain. Compared with retraining from stratch (around 1100s), it takes only about 5% to obtain an R-(0.3,0.2) guidance model (around 40s). With efficient guidance model obtaining, the overall FF-Erase unlearning cost is significantly lower than retraining while forgetting as effective as retraining, as shown in §6 and §C.

## D USAGE OF LLM

In this paper, we use the Large Language Model (LLM) to aid or polish writing. Details are described as follows:

In the interest of full transparency, we utilized LLMs (including Gemini-2.5-pro and GPT-5) as a writing assistant to enhance the clarity, conciseness, and overall readability of this manuscript. The LLM's role was strictly limited to improving sentence structure, grammar, and flow. All scientific contributions, experimental results, and core arguments were conceived and articulated exclusively by the authors, who reviewed and approved every revision to ensure the integrity and accuracy of the final text.

