# OpenReview forum: "FF-Erase : Machine Unlearning and Verification for Forward-Forward Models"
_ICLR.cc/2026/Conference — Submitted to ICLR 2026_

### Official Review · Reviewer_y2EG · 2025-10-28

**Soundness:** 3
**Presentation:** 3
**Contribution:** 3
**Rating:** 6
**Confidence:** 2

**Summary:**

This paper presents the first unlearning method for forward-forward models as well as a membership inference attack for the FF architecture.

**Strengths:**

Rethinking machine unlearning for models that do not use backpropagation is highly interesting, as no prior work exists and discoveries might lead to advances in both areas (BP and FF).

The new method is intuitive (similar to the popular SCRUB method in BP unlearning) and shows good performance in the experiments.

The G-MIA also makes sense, adapting traditional MIA to the layer level of FF models.

Both the new method and G-MIA are adapting existing working approaches to the FF setting, providing a valuable contribution for future research.

To preempt this point from other reviewers: Both FF-Erase and G-MIA are adaptations to proven approaches in BP unlearning. While some might call this a lack of novelty, I see the adaptation to FF as non-trivial and valuable.

**Weaknesses:**

The literature is missing BP unlearning methods that also utilize student-teacher approaches (SCRUB, Bad Teacher, …). Adding this literature and highlighting why it fails (ideally with experiments) would improve the paper.

**Questions:**

Why not use the original model as the guidance model and perform unlearning like SCRUB? I may have missed it, but the rationale behind training a new guidance model instead of using the existing starting model eludes me (it has been shown to work in BP unlearning).

---

> ### Author Response · Authors · 2025-11-19
> **To Reviewer y2EG:**
>
> We sincerely thank the reviewer for the recognition of our work and for the constructive comments that helped us improve the manuscript. Let us answer the weaknesses and questions one by one.
>
> **Weakness:** Thank you for your valuable suggestion. In response, we have added a literature review of these approaches in Appendix A and highlighted why they fail with experiments in Appendix C.3.
>
> Those distillation-based GA methods utilize the final-layer logits for removing specific data's influence from the original model, which ignores the residual knowledge that persists in the previous layers of an FF model and leads to ineffective unlearning as GA (using small $\lambda$) does. To be specific, SCRUB achieves successful unlearning in the BP model by encouraging the unlearned model $\theta^u$ to perform similarly to the original model $\theta^o$ on remaining data, while encouraging $\theta^u$ to perform differently from $\theta^o$ on forgetting data. However, for FF models, directly encouraging $\theta^u$ to move away from $\theta^o$ does not guarantee a valid goodness distribution, risking the model collapse as all gradient-ascent-based methods do. Bad Teacher is another famous distillation-based approximate unlearning approach, which is also infeasible to directly adapt to FF models. This is because it depends on the final-layer logits for unlearning, which is not enough, as FF models also persist knowledge in their previous layers during their layer-wise greedy learning.
>
> We have also added experiments using those distillation-based unlearning methods in Appendix Section C.3. The experimental results show that the unlearned model using the Bad Teacher method maintains obviously high accuracy on the remaining data, indicating ineffective unlearning. Moreover, it could not pass the G-MIA test, and the CKA scores shown in Appendix C.2 also demonstrate that Bad Teacher can not remove residual knowledge in the previous layers. The experimental results are consistent with our previous conclusion. Thank you again for your valuable comments.
>
> **Question:** As we explained in answering the weakness, SCRUB uses the original model as the guidance model and unlearns by *moving away from correct logits*. However, FF-Erase requires *shifting towards the ignorant goodness distribution*, since moving away from correct goodness can lead to an away-but-invalid goodness distribution for FF models. Such an invalid goodness distribution can lead to model collapse due to FF models' unique challenge of sensitivity in parameter tuning.
> To conclude, due to FF models' unique challenges, it is necessary to additionally generate a guidance model rather than directly using the original model.

---

> > ### Comment · Reviewer_y2EG · 2025-11-26
> >
> > Thank you very much for your clarification. I am still leaning towards accepting, and am mainly hindered in increasing (or decreasing) my score by my limited knowledge in FF as reflected by my confidence score. Interesting work!

---

### Official Review · Reviewer_4v3L · 2025-10-28

**Soundness:** 2
**Presentation:** 2
**Contribution:** 2
**Rating:** 4
**Confidence:** 3

**Summary:**

The paper presents an unlearning algorithm that is tailored for Forward-Forward network where traditional unlearning methods may cause model collapse. The proposed method follows the spirit of the common unlearning strategy that preserves retain datasets performance (or goodness here) and reduce the forget set's performance. The loss function for reducing forget set's goodness in this paper is by reducing its difference to a guidance model with KL divergence. The results show the proposed method works in terms of against MIA.

**Strengths:**

1. The problem addressed in this paper is quite unique. The Forward-Forward (FF) network is not yet widely adopted in practice, and it is unclear why one would need to consider data unlearning for such models at this stage. Nevertheless, the paper takes an interesting forward-looking perspective by identifying and tackling a potential issue that others have not yet considered.

**Weaknesses:**

1. The paper is not well polished and contains several errors. For example, $\mathbf{g}^l$ is defined as a vector with a dimensionality equal to the number of classes, yet the paper also states that $\mathbf{g}^l = ||\mathbf{h}^l||_1$. The $L_1$ norm of a vector is a scalar, not another vector, which raises confusion about how this formulation is meant to work. What is the correct equation here?

2. The method rely on a guidance model to learn how to behave on forget set. The funny thing is that if one can have a guidance model that is ignorant of the forgetting data, why do we need to do the unlearning? In addition, the training of guidance model through equation 7-8 shows it is not a trivial task. It makes me wander the whole purpose of making up this problem. One should retrain the FF model from scratch, which might be cheaper?

3. The paper claims that conventional unlearning methods can cause model collapse in Forward-Forward networks, but no empirical evidence or citation is provided to support this claim. It remains unclear whether this is based on actual observation or merely intuition. Providing experimental or theoretical justification would strengthen the paper’s motivation.

**Questions:**

My questions are listed in the weakness section

---

> ### Author Response · Authors · 2025-11-19
> **To Reviewer 4v3L (1/2):**
>
> Thank you for your comments and questions. We would like to provide you with a more detailed explanation and clarification, and we hope our answer to your questions can address your concerns.
>
> **For weakness 1:** We apologize for the confusion and would like to clarify that the notations and equations are actually correct, which follow the state-of-the-art FF training algorithm (Equation (4-8) in DeeperForward [1], ICLR'25). Using contrastive FF learning [2], each layer generates $J$ outputs for every class, where $J$ is the number of classes. Therefore, the $\mathcal{h}^l$ is actually a vector of vectors, which can also be understood as a matrix with the shape of $(d_l, J)$, where $d_l$ is the dimension of the $l$-th layer's output. Consequently, when we apply *column-wise L1 norm* on a vector of vectors $\mathcal{h}^l$, we actually get a vector of scalars $\mathcal{g}^l$. We have added more explanations in Section 3.1 in the updated paper to avoid confusion.
>
> [1] SUN L, ZHANG Y, HE W, et al. DeeperForward: Enhanced Forward-Forward Training for Deeper and Better Performance[C]//The Thirteenth International Conference on Learning Representations. 2025.
>
> [2] PAPACHRISTODOULOU A, KYRKOU C, TIMOTHEOU S, et al. Convolutional channel-wise competitive learning for the forward-forward algorithm[C]//Proceedings of the AAAI Conference on Artificial Intelligence. 2024, 38(13): 14536-14544.
>
> **Weakness 2:** Thank you for your question. FF-Erase using the guidance model still shows a significantly better performance than retraining the FF model. We would like to answer this question in the following three points:
>
> Firstly, the guidance model has poor performance on the test set and the remaining data, although it is also ignorant of the forgetting data. For example, in Figure 4, the guidance model only has an accuracy of 58.95% on forgetting data and 61.55% on remaining data, which is not capable of serving as an unlearned model. We show the guidance model's performance on the forget and remain data (CIFAR10 VGG13) as follows:
>
> | Training Time | 83.7s | 158.9s | 234.4s | 385.0s | 912.2s |
> | --- | --- | --- | ---| --- |  --- |
> | Acc on D_forget | 52.28% | 58.95% | 63.98% | 66.26% | 72.54% |
> | Acc on D_remain | 53.49% | 61.55% | 67.74% | 69.78% | 76.98% |
>
> We show the retraining model's performance on the forget and remain data (CIFAR10 VGG13) for comparison as follows:
>
> | Training Time | 68.9s | 157.4s | 239.6.4s | 388.2s | 906.2s |
> | --- | --- | --- | ---| --- |  --- |
> | Acc on D_forget | 63.54% | 74.35% | 77.02% | 79.54% | 82.04% |
> | Acc on D_remain | 67.53% | 81.69% | 86.44% | 90.28% | 92.69% |
>
> Compared with the performance on both forgetting data and remaining data of the golden standard retraining method, the guidance model has a much lower accuracy on both forget and remain data, regardless of the training time, and thus is not suitable to directly serve as the unlearned model.
>
> Secondly, generating a guidance model is actually quite rapid. As we introduced in Equation (7-8) and Section 4.2, we only utilize a small subset of the remaining data and stop retraining/distilling quite earlier than complete retraining/distilling. Evaluation results in Figure 4(a) and Figure 4(b) also demonstrate that, compared with the retraining method (RE), generating a guidance model (dashed lines in FF methods) requires significantly less time.
>
> Thirdly, FF-Erase using the guidance model is still cheaper than retraining. Our experimental results in Table 1 report the total unlearning time consisting of two parts: generating the guidance model and tuning the parameters of the original model. We can see that even using (60.7-410.5 s) for generating the guidance model, the overall FF-Erase unlearning time (353.7-583.5 s) is still much efficient than the retraining time (around 1100s as shown in Figure 4), while acquiring an unlearned model with excellent performance compared to the retraining method.
>
> To conclude, the guidance model is not feasible as an unlearned model. It is fast to acquire and assists FF-Erase to overcome the inherent challenges introduced by FF unlearning. Overall, FF-Erase unlearning cost, containing the guidance model generating time, is still much cheaper than retraining while achieving similar unlearning performance.

---

> ### Author Response · Authors · 2025-11-19
> **To Reviewer 4v3L (2/2):**
>
> **Weakness 3:** Thanks for your comment. We would like to clarify that we have provided a comprehensive experimental result as the empirical evidence in Section 6.2, 6.3, and Appendix C for the phenomenon, that FF models are more sensitive to parameter tuning than BP models, and conventional approximate unlearning methods lead to either model collapse or ineffective unlearning.
>
> Firstly, in Section 6.2, the pink line in Figure 4 shows the results of conventional approximate unlearning methods (using direct gradient ascent (GA), as the representative) on FF models, which leads to an obvious model collapse: the test accuracy keeps an obviously low level and can no longer increase.
>
> Secondly, in Section 6.3, we take a whole subSection to further explore different settings ($\lambda$=10^1,10^0,10^{-1},10^{-2},10^{-3},0) of GA for a comprehensive demonstration. The result indicates that the model either collapses or can not unlearn the forgetting data (when the $\lambda$ is too small, the gradient ascent can only introduce a negligible impact on the model).
>
> Thirdly, we have conducted extra experiments to show other conventional unlearning methods as updated in Appendix Section C.3. The new baselines include advanced gradient ascent methods, distillation-based methods, and advanced retraining methods. The new result is aligned with our conclusion: retraining methods are still not efficient enough; approximate unlearning methods either lead to model collapse or unlearn ineffectively.
>
> Thank you again for your comments, and we will be happy to address any remaining concerns.

---

> > ### Comment · Reviewer_4v3L · 2025-11-26
> >
> > Thanks for addressing my previous concerns.
> >
> > For weak point 2, I am still not very convinced. I don't know how to explain "guidance model has poor performance on the test set and the remaining data" as an legit reason of inventing this method. It doesn't directly answer my "what-if" question directly.  Also, while I understand the method is "more efficient" than retraining in the experiments, training guidance model is still non-trivial effort as equations stated in the paper as stated in the original comments. We can arbitrarily reduce/increase the data size to get different observation in terms of wall-clock computation time; but the method requires to train an additional model from scratch/distillation. It needs additional operation time and computation graph loading resources etc. Is it still practical? Under which condition set the method is better or worse than retraining?

---

> > > ### Author Response · Authors · 2025-11-27
> > > **Reply to Reviewer 4v3L's response (1/2):**
> > >
> > > Thank you for your continued engagement and for giving us the opportunity to further clarify our work. We are glad to know that the previous concerns have been addressed. We understand your remaining concerns regarding the guidance model and would like to address them directly. Let us answer your questions one by one.
> > >
> > > > I don't know how to explain "guidance model has poor performance on the test set and the remaining data" as an legit reason of inventing this method.
> > >
> > > * We would like to clarify: it is that we make use of a guidance model to assist *goodness decrease*, rather than that we invent FF-Erase due to a failure of guidance models.
> > >     * **Guidance models do not directly address the unique challenges of FF unlearning.** FF-Erase begins with exploring a practical (*effective*, *efficient*, and unlearned model *utility*) unlearning method for FF models. After observing the failure of existing BP unlearning methods, we realize the unique challenges of the FF scenario, and thus propose FF-Erase to unlearn via *goodness decrease*. Our major contribution is proposing the first FF unlearning method by FF-Erase's novel goodness-guided framework, which addresses the challenges. The guidance models are neither designed for convergence nor designed with specific algorithms to deal with FF models' layer-wise learning and FF models' sensitive parameter to tuning, thus do not work.
> > >     * **The guidance model is not an alias of a retrained model.** We give two instances of generating guidance models, which do not indicate that guidance models are fixed by those two strategies. Any ignorant model (no other requirements like *model utility*) with the same shape as the original FF model can serve as a guidance model.
> > >     * **It is actually convenient to generate a guidance model.** As illustrated in Section 4.2, we only require the ignorance of the guidance model. Consequently, the objective of guidance models is totally different from retraining models or distillation models. In another world, we do **not** insist on achieving an optimal of minimizing the loss on retraining data (as we never write). The process described in Equation (7) or (8) aims at only ignorant models.
> > >
> > > >  Also, while I understand the method is "more efficient" than retraining in the experiments, training guidance model is still non-trivial effort as equations stated in the paper as stated in the original comments. We can arbitrarily reduce/increase the data size to get different observation in terms of wall-clock computation time; but the method requires to train an additional model from scratch/distillation.
> > >
> > > >  It needs additional operation time and computation graph loading resources etc. Is it still practical?
> > >
> > > We would like to apologize for potential misunderstanding here if our Equations (7) and (8) mislead you about some objectives, like minimizing the loss on retraining data.
> > >
> > > * We would like to clarify that the training guidance model is actually convenient and requires a moderate effort:
> > >     * Firstly, **it is really rapid**. Since we only require the ignorance of the guidance model, we can use a smaller proportion $\alpha_1$ of retraining/distilling data to obtain a reasonably good guidance model, as you agreed later (*...reduce/increase the data size to get different observation in terms of wall-clock computation time...*). Experimental results in Sections 6.2 and 6.4 show that training guidance models take ~40-150s while complete retraining requires ~1170s. This benefit is attributed to the fundamental difference in objectives between guidance models (ignorance) and retrained/distilled models (convergence and utility).
> > >     * Secondly, **it needs few extra operations**. The guidance models share the same architecture as the original model. In practice, this means we reuse the code path and hardware setup with only a different random seed and a filtered dataloader. No additional framework engineering is required.
> > >     * Thirdly, **it does not occupy more memory resources**. The guidance models are released after unlearning immediately, introducing no extra memory overhead. Even the peak memory usage with using the guidance model is 2 times as retraining, which is acceptable in practice (existing unlearning methods for BP models require much larger space; retraining methods [1-3] store O(n) intermediate parameters where n is the number of training rounds, distillation-based approximate unlearning [4,5] requires at least 3 times as well).
> > >
> > > [1] LIU G, MA X, YANG Y, et al. Federaser: Enabling efficient client-level data removal from federated learning models[C]//2021 IEEE/ACM 29th International Symposium on Quality of Service (IWQOS). 2021: 1-10.
> > >
> > > [2] TAO Y, WANG C, PAN M, et al. Communication-efficient and provable federated unlearning[J]. Proceedings of the VLDB Endowment, 2024, 17(5): 1119-1131.

---

> > > ### Author Response · Authors · 2025-11-27
> > > **Reply to Reviewer 4v3L's response (2/2):**
> > >
> > > [3] XIONG Z, LI W, LI Y, et al. Exact-fun: an exact and efficient federated unlearning approach[C]//2023 IEEE International Conference on Data Mining (ICDM). 2023: 1439-1444.
> > >
> > > [4] KURMANJI M, TRIANTAFILLOU P, HAYES J, et al. Towards unbounded machine unlearning[J]. Advances in Neural Information Processing Systems, 2023, 36: 1957-1987.
> > >
> > > [5] CHUNDAWAT V S, TARUN A K, MANDAL M, et al. Can bad teaching induce forgetting? unlearning in deep networks using an incompetent teacher[C]//Proceedings of the AAAI Conference on Artificial Intelligence. 2023, 37(6): 7210-7217.
> > >
> > >
> > > > It doesn't directly answer my "what-if" question directly.
> > >
> > > > (if one can have a guidance model that is ignorant of the forgetting data, why do we need to do the unlearning?)
> > >
> > > * **If one have a guidance model**, they still need unlearning. Because the guidance model is not designed for addressing FF models' unique issues in unlearning. Directly using the guidance model as an unlearned model leads to ineffective/inefficient/poorly-performed unlearning. (Having a guidance model is not equivalent to having a retrained model.)
> > >
> > > > Under which condition set the method is better or worse than retraining?
> > >
> > > * In our extensive experiments, FF-Erase is always much **more efficient (cheaper)** while achieving **effective unlearning** and maintaining the unlearned model's **utility** as retraining does.
> > > * As the experimental results demonstrated in Section 6 and Appendix C, FF-Erase always achieves better efficiency than retraining, compared between ***FF-Erase's overall time*** (including generating guidance models and tuning parameters to shift the goodness distribution) and ***retraining time***. Meanwhile, FF-Erase achieves effective unlearning as retraining does, measured by accuracy on forgetting data, Centered Kernel Alignment (CKA) scores, and G-MIA accuracy. Moreover, FF-Erase maintains the unlearned model's utility as retraining does, compared to the accuracy on test data.
> > >
> > > Thank you again for your thoughtful feedback and for giving us the opportunity to address your concerns in detail. We faithfully hope this explanation alleviates your concerns and demonstrates the practical value of FF-Erase. We are happy to incorporate any further suggestions or questions if you have.

---

### Official Review · Reviewer_r4jW · 2025-11-01

**Soundness:** 2
**Presentation:** 3
**Contribution:** 3
**Rating:** 4
**Confidence:** 3

**Summary:**

Forward-forward (FF) models provide an alternative to backpropagation algorithms. Machine unlearning for FF models, which aims to remove specific data knowledge from a model, has not been explored. Using classic unlearning algorithms can cause model collapse on these models. To fill this gap, the paper introduces FF-Erase, a novel machine learning algorithm tailored to FF models. The method uses a guidance model (obtained via mini-retrain or fast distillation) to provide goodness scores and steer the original model. The authors also propose G-MIA, a goodness-based membership inference attack for evaluation. Experiments on CIFAR/MNIST variants show that FF-Erase achieves similar unlearning effectiveness to retraining while being faster.

**Strengths:**

Strengths:
* The motivations and presentation are clear. The authors provides a novel approach for machine unlearning in FF models, which has not been explored in the literature.

* The authors also introduce a novel goodness-based MIA metric, and show its effectiveness.

**Weaknesses:**

* The proposed method is mainly compared with GA and retraining as baselines. Can the performance of other unlearning methods be assessed?

* The paper states that applying existing unlearning updates causes model collapse in FF models, but the explanation is mostly high level. However, more thorough analysis as to why this is the case in FF models would be helpful.

* The method requires obtaining a guidance model, which causes an initial overhead. It also requires maintaining both the original and guide model, and forward passes through both models.

* The guidance model is distilled from the original model, and is then used to guide the original model through a KL divergence loss, which could be inefficient.

Minor issues:

* Some colors in the figure 5(a) and (b) can be hard to distinguish (lambda=1, 0.1)
* What the dashed lines represent in figure 4(a) and (b) is not clear.

**Questions:**

* What is the performance of  the guidance model on the forget and remain data?

---

> ### Author Response · Authors · 2025-11-19
> **To Reviewer r4jW (1/2):**
>
> Thank you very much for your careful review and detailed questions. We apologize for the possible misunderstanding, and we would like to show extra experimental results and give a more concrete explanation.
>
> **Weakness 1:** Yes, of course. **The performance (regarding efficiency, effectiveness, and utility) of other common BP unlearning methods on FF models is added in Appendix Section C.3**, where we add baselines including: an advanced retraining method (FATS) and three state-of-the-art approximate BP unlearning methods: Fast-yet-effective (FYE), Bad Teacher (BT), and SURE. FATS is a state-of-the-art and representative retraining-based method, which is proposed for the FL scenario but also achieves state-of-the-art performance for centralized methods. FYE is a straightforward unlearning method based on gradient ascent, which is still one of the most efficient approximate unlearning methods. We additionally pick up a representative method, Bat-teacher, which utilizes distillation for approximate unlearning, as suggested by other reviewers.
>
> The experimental result is aligned with our conclusion in Section 1 and Section 2, that the retraining-based methods are still not efficient enough, while the approximate unlearning methods lead to either model collapse or ineffective unlearning.
>
> **Weakness 2:** We realize and verify this phenomenon through comprehensive experiments, including various models (TinyCNN/AlexNet/VGG) and datasets (MNIST/FMNIST/CIFAR10/CIFAR100) as shown in Section 6 and Appendix C. The rationale behind this phenomenon is that FF algorithms use greedy and layer-wise training approaches to replace backpropagation, so that each layer is independently optimized on its local goodness objective. Therefore, parameters in the previous layers do not strictly update towards a consistent direction with the subsequent layers, nor compress everything “useful” for the final output layer. Consequently, FF models exhibit heightened sensitivity to parameter tuning due to their BP-free nature, making existing BP unlearning methods not feasible to be directly adapted.
>
> Since this is the first paper on machine unlearning for FF models, we focus on proposing a specific unlearning and verification method for FF models. We will leave the theoretical analysis of why existing work leads to model collapse to future work.
>
> **Weaknesses 3 and 4:** The extra overhead introduced by guidance model design from FF-Erase is acceptable in terms of time complexity and space complexity.
>
> First, for the time complexity/efficiency, even with the time for obtaining the guidance model, the overall unlearning time of FF-Erase is still less than state-of-the-art methods effective for FF unlearning (i.e., retraining and retraining-based methods). FF-Earse is 1.9-3.1 times faster than retraining and FATS, as shown in Section 6.3 and Appendix C.3. State-of-the-art approximate BP unlearning is not feasible for FF models, as the details given in Appendix C.2.
>
> Secondly, for the space complexity, FF-Erase takes at most 2 times the memory/disk space compared with the FF training process, with the space complexity of O(1). In contrast, existing unlearning methods for BP models require much larger space (retraining methods [1-3] store O(n) intermediate parameters where n is the number of training rounds, distillation-based approximate unlearning [4,5] requires at least 3 times as well). Moreover, FF models are inherently efficient in memory consumption compared to BP models (when training, BP requires storing all activated neurons from all layers, while FF only stores neurons from one layer). An extra one-time memory consumption is tolerable.
>
> [1] LIU G, MA X, YANG Y, et al. Federaser: Enabling efficient client-level data removal from federated learning models[C]//2021 IEEE/ACM 29th International Symposium on Quality of Service (IWQOS). 2021: 1-10.
>
> [2] TAO Y, WANG C, PAN M, et al. Communication-efficient and provable federated unlearning[J]. Proceedings of the VLDB Endowment, 2024, 17(5): 1119-1131.
>
> [3] XIONG Z, LI W, LI Y, et al. Exact-fun: an exact and efficient federated unlearning approach[C]//2023 IEEE International Conference on Data Mining (ICDM). 2023: 1439-1444.
>
> [4] KURMANJI M, TRIANTAFILLOU P, HAYES J, et al. Towards unbounded machine unlearning[J]. Advances in Neural Information Processing Systems, 2023, 36: 1957-1987.
>
> [5] CHUNDAWAT V S, TARUN A K, MANDAL M, et al. Can bad teaching induce forgetting? unlearning in deep networks using an incompetent teacher[C]//Proceedings of the AAAI Conference on Artificial Intelligence. 2023, 37(6): 7210-7217.

---

> ### Author Response · Authors · 2025-11-19
> **To Reviewer r4jW (2/2):**
>
> **Question:** We show the guidance model's performance on the forget and remain data (CIFAR10 VGG13) as follows:
>
> | Training Time | 83.7s | 158.9s | 234.4s | 385.0s | 912.2s |
> | --- | --- | --- | ---| --- |  --- |
> | Acc on D_forget | 52.28% | 58.95% | 63.98% | 66.26% | 72.54% |
> | Acc on D_remain | 53.49% | 61.55% | 67.74% | 69.78% | 76.98% |
>
> We show the retraining model's performance on the forget and remain data (CIFAR10 VGG13) for comparison as follows:
>
> | Training Time | 68.9s | 157.4s | 239.6.4s | 388.2s | 906.2s |
> | --- | --- | --- | ---| --- |  --- |
> | Acc on D_forget | 63.54% | 74.35% | 77.02% | 79.54% | 82.04% |
> | Acc on D_remain | 67.53% | 81.69% | 86.44% | 90.28% | 92.69% |
>
> Compared with the performance on both forgetting data and remaining data of the golden standard retraining method, the guidance model has a much lower accuracy on both forget and remain data, regardless of the training time, and thus is not suitable to directly serve as the unlearned model. Note that in FF-Erase, a short training time is enough for obtaining the guidance model (e.g., the distillation of the guidance model stops at 83.7s as shown in D-(0.5,0.1) in Table 1).
>
> Based on the provided experimental results, we would like to clarify three specific points about FF-Erase. Firstly, directly distilling on the remaining data for unlearning is not efficient for FF models. Secondly, FF-Erase does not employ distillation for complete unlearning. It stops distilling at a very early stage, only to acquire an ``ignorant'' guidance model. Thirdly, the overall unlearning time (both generating the guidance model and tuning parameters), as we showed in Figure 4, is significantly efficient compared with baselines. We have added more explanations regarding the guidance model in Appendix Section C.4 in the updated paper.
>
> **Minor issues 1:** We are grateful for your kind reminder, and we have picked up better color choices for a clearer presentation in Figure 5.
>
> **Minor issues 2:** We apologize for the confusion, which might lead to your concerns in Weakness 4. In Figures 4(a) and 4(b), we showed the overall unlearning time, consisting of the time for generating the guidance model (using dashed lines) and the time for tuning the parameters (using solid lines). Therefore, the accuracy of dashed lines starts from 10% because we are generating the guidance model. The accuracy for the solid lines starts from about 90% and decreases gradually because the parameter tuning starts from the original model. Thank you again for pointing this out, and we have improved the clarity in the updated paper.

---

> ### Author Response · Authors · 2025-11-27
>
> We really appreciate the constructive feedback and the time you have taken to review our paper. As the discussion period deadline is approaching, we would like to respectfully remind you to consider our responses. We hope our revisions have addressed your concerns, and we are glad to engage in further discussion if needed.
>
> If the revisions and the responses meet your expectations, we would like to kindly request you to reconsider raising your rating score to reflect the improvements made to the paper. Your evaluation plays a crucial role in the visibility and impact of FF-Erase. We sincerely extend our gratitude to you for your time and thoughtful suggestions.

---

### Official Review · Reviewer_CRuv · 2025-11-01

**Soundness:** 3
**Presentation:** 3
**Contribution:** 4
**Rating:** 8
**Confidence:** 3

**Summary:**

Here, the authors explore the critical challenge of machine unlearning with the Forward-Forward networks. The authors propose FF-ERASE to efficiently remove the influence of specific training data from FF models. The method tackles the unique difficulties posed by FF architectures, such as their sensitivity to parameter changes and layer-wise greedy optimization, which can easily trigger catastrophic model collapse during the unlearning process.

**Strengths:**

- It is the first formalized solution for machine unlearning in the increasingly relevant and research-intensive FF network paradigm, filling a critical gap.
- The method is tailored to the layer-wise optimization of FF models, suggesting better unlearning method than adapting a BP-based unlearning technique.

- Novel method which directly tackles the significant practical challenge of catastrophic model collapse that FF models face when their parameters are forcefully adjusted, making the unlearning process feasible.

**Weaknesses:**

- The core principle of FF is layer-wise, greedy optimization, which means there is no global loss signal tying all layers together. When unlearning, removing knowledge from one layer (e.g., a shallow layer's feature representation) doesn't guarantee that the influence is fully pruned in deeper layers. This lack of global coherence means residual knowledge traces of the forgotten data might persist in high-level feature spaces, making the unlearning reversible, even if the final classification decision is erased. Use Centered Kernel Alignment (CKA) or a similar representation similarity metric to compare the feature spaces generated by $D_{\text{forget}}$ samples in three different layers (shallow, middle, deep) before and after FF-ERASE.

- Computation efficiency and complexity of unlearning in FF models are not compared to the BP based unlearning methods.
Directly compare the total computational time required for FF-ERASE to unlearn a specified set $D_{\text{forget}}$ against the most efficient BP-based unlearning methods (e.g Gradient Ascent ) on a CNN model.

**Questions:**

See Weakness

---

> ### Author Response · Authors · 2025-11-19
> **To reviewer CRuv (1/2):**
>
> First of all, we really appreciate your insightful comments and valuable feedback. We have added CKA-related experiments and more efficiency comparisons to BP-based unlearning methods to better address your concerns. Let us answer your questions one by one.
>
> **Weakness 1:** It is true that removing knowledge from one layer is not enough for FF models due to their layer-wise and greedy optimization. However, FF-Erase can prevent residual knowledge traces from all layers. According to your suggestion, we show the Centered Kernel Alignment score of each layer between the original model and the unlearned model when forgetting data forwards the network as follows:
>
> | Layer | 1 | 2 | 3 | 4 | 5 |
> |---|---|---|---|---|---|
> | RE | 0.9918 | 0.9249 | 0.8004 |0.6319 |0.6286 |0.5700 |0.3619|
> | FF-Erase(D) | 0.9958 | 0.9658 | 0.9066 | 0.8100 | 0.7584 |
> | FF-Erase(R) | 0.9961 | 0.9721 | 0.9199 | 0.8279 | 0.7816 |
>
> | Layer | 6 | 7 | 8 | 9 | 10 |
> |---|---|---|---|---|---|
> | RE | 0.5700 | 0.3619 | 0.3472 | 0.4981 | 0.4200 |
> | FF-Erase(D) | 0.7309 | 0.6168 | 0.5737 | 0.5462 | 0.4548 |
> | FF-Erase(R) | 0.7454 | 0.6292 | 0.5919 | 0.5930 | 0.5128 |
>
> | Layer | 11 | 12 | 13 |
> |---|---|---|---|
> | RE | 0.4923 | 0.6223 | 0.6144 |
> | FF-Erase(D) | 0.5071 | 0.5742 | 0.5488 |
> | FF-Erase(R) | 0.5748 |0.6490 | 0.6358 |
>
> This CKA score corresponds with the experiment shown in Figure 4 in Section 6.2, where we consider an FF VGG13 model using CIFAR10. RE means the CKA score between the retraining model (RE) and the original model, which serves as a standard of complete unlearning. A CKA score significantly higher than RE indicates that this layer may persist residual knowledge (ineffective unlearning), while a CKA score significantly lower than RE indicates that the representations from this layer are pushed away from the original model (potential over-forgetting). An effective unlearning algorithm generates an unlearned model with a similar CKA score to RE layer-wise.
>
> According to layer-wise CKA scores, FF-Erase (guided by both fast-distilled (D) and mini-retrained (R) guidance models) effectively removes the residual knowledge from all layers.
>
> To better address your concerns, **we specifically added a section, Appendix C.2, in the paper to compare the CKA scores of all baseline unlearning methods in detail**. We really hope you can review this Section for more CKA-related details. ***Experimental results show that our FF-Erase achieves CKA scores very similar to retraining methods, while other approximate unlearning methods are either ineffective or degrade the utility of unlearned FF models.***
>
> There is a shortcoming for CKA and goodness vector when verifying whether the unlearning is thorough or not: we need to determine a standard, e.g., here we use the CKA scores from retraining as the standard. This can be challenging sometimes, for example, if the remaining data holds enough features and the leaving data is not unique, then the unlearned model (even a vanilla retrained model) actually has a small CKA score.
>
> To better handle this challenge, we propose Goodness Membership Inference Attack (G-MIA) besides CKA, which is also an important contribution of this paper. As introduced in Section 5, G-MIA takes every layer's goodness as input, employs an MLP, and outputs whether a test sample is a member or not. Given an unlearned model, if G-MIA can not distinguish the forgetting data from non-member data, the unlearning algorithm is considered effective. If residual knowledge persists at any (no matter shallow, intermediate, or deep) layer, G-MIA will capture such a signal and distinguish the forgetting data from non-member data. By such an attack, we address the challenge to accurately verify FF unlearning algorithms without depending on a threshold.

---

> ### Author Response · Authors · 2025-11-19
> **To reviewer CRuv (2/2):**
>
> **Weakness 2:** We would like to clarify that we have compared the total unlearning time of FF-Erase (including generating guidance models and tuning for goodness decrease) with your required efficient BP unlearning methods (gradient ascent), as shown in Figure 4 (a) and Figure 4 (b) in Section 6.2. The yellow and orange lines (consisting of dashed lines part and solid lines part) refer to FF-Erase guided by fast-distilled (D) and mini-retrained (R) guidance models, respectively. The dashed lines refer to the accuracy of the guidance model. For example, in FF-Erase(R), it takes around 70s for generating a guidance model, and it takes another around 260s for FF-Erase unlearning, represented by the solid line that follows the dashed line in the same color. Therefore, the total/overall/end-to-end unlearning time is around 330s. The pink line (consisting of only a solid line part because direct gradient ascent does not require generating a guidance model) presents for BP GA unlearning methods, which leads to model collapse after about 70s. Therefore, Figure 4 shows the total computational time for FF-Erase and Gradient Ascent. We apologize for not clearly explaining the figure in the paper, and we have updated the captions for a more precise description. We also showed the total computational time for FF-Erase and GA under different settings (CIFAR10/CIFAR100/MNIST/FMNIST using VGG113/VGG16/TinyCNN/AlexNet) in Appendix Section C.1.
>
> **We also compare more state-of-the-art efficient BP methods (robust GA, distillation-based GA, and retraining methods) in Figure 10 in Appendix Section C.3.** The experiment results show that for effective FF unlearning (neither ineffective nor leading to model collapse), FF-Erase is the most efficient one.
>
> Thank you again for your insightful comments and valuable suggestions! We will be happy to address any remaining concerns (if any).

---

### Author Response · Authors · 2025-11-24
**Summary of Rebuttal**

Dear Area Chair and Reviewers,

We sincerely thank you for your time and for providing insightful and constructive feedback on our manuscript. We have embraced your suggestions and, in response, have undertaken substantial revisions to the paper. These include new experiments, an expanded comparative analysis, and clarifications, which we are confident that we have significantly strengthened our work and addressed all raised concerns.

Below is a summary of the major revisions, cross-referenced to the specific reviewer comments they address. Detailed point-by-point responses are provided in the full rebuttal.

* **A more comprehensive literature review** in *Appendix A*. We have significantly expanded the literature review and provided more details on why existing approximate unlearning methods for BP models fail at the FF scenario, including state-of-the-art distillation-based gradient ascent (teacher-student approaches) and other state-of-the-art robust gradient ascent methods.
* **A supplementary experiment** leveraging the Centered Kernal Align (CKA) method to verify the effectiveness of unlearning layer-wise, which is shown in *Appendix C.2*. We show the CKA scores layer-wise and demonstrate that FF-Erase achieves effective unlearning for all (shallow, middle, and deep) layers.
* **Improving the experiment** by adding more baselines, which is shown in *Appendix C.3*. We add a state-of-the-art exact BP unlearning method and three state-of-the-art approximate unlearning methods as new baselines. The results from improved experiment settings are consistent with our conclusion in the main text, that existing BP unlearning methods are not suitable for FF models, and our FF-Erase is the first effective and efficient FF unlearning method.
* **Showing supplementary experimental result** in *Appendix C.4*. We show a complete training curve of the guidance models, demonstrating that they have a much lower accuracy on both forget and remain data, regardless of the training time, and thus are not suitable to directly serve as the unlearned model.
* **Revised figure caption** for *Figure 4 in Section 6.2* and **imporving figure color** for *Figure 5 in Section 6.3*. We explain the meaning of dashed lines and which lines are represented in the guidance model. We also improve the colors to be more distinguishable.
* **Clarifications to address some misunderstandings** in rebuttal. We clarify that the guidance models are not capable of serving as unlearning models, but it is efficient to acquire and FF-Erase's overall unlearning time (including obtaining a guidance model) is still significantly less than retraining. We also clarify that our notations and equations are correct.

We are confident about those responses and hope these substantial improvements have significantly strengthened the paper's contributions. We are grateful for the opportunity to improve our work based on your valuable feedback. We hope our revisions and detailed rebuttal have fully addressed your concerns, and we will be happy to engage in further discussion. We kindly invite the reviewers to consider the enhanced quality of our manuscript in their final assessment.

Thank you again for your valuable time and consideration.

---

### Meta-Review · Area_Chair_QxFT · 2026-01-07

**Summary:**

FF-Erase proposes the first dedicated machine unlearning method for Forward-Forward (FF) models, a training paradigm that differs fundamentally from backpropagation. The method uses an “ignorant” guidance model and a KL-based goodness alignment objective to erase information associated with a forget set, while periodically running a recovery step on retained data to prevent collapse.

**Reviewer Concerns:**

Reviewer CRuv:

(1) Layer-wise residual knowledge might persist; asked for CKA (or similar) across shallow/mid/deep layers — Addressed. Authors added layer-wise CKA analysis and report CKA scores per layer, aligning FF-Erase to the retraining “standard,” and say Appendix C.2 contains full baseline comparisons.

(2) Minor clarity issues about compute times -- addressed.

Reviewer r4jW:

(1) Baselines mainly GA + retraining; asked to assess other unlearning methods — Addressed. Authors add FATS + FYE + Bad Teacher + SURE and claim these either collapse or are ineffective for FF; they point to Appendix C.3.

(2) “Why do BP unlearning updates collapse FF models?” explanation was high-level; asked for deeper analysis — Largely unaddressed... Authors give a mechanistic explanation tied to FF’s layer-wise greedy objectives and sensitivity, but explicitly defer theoretical analysis to future work.

(3) Guidance model overhead / needing to maintain two models / KL guidance might be inefficient — Largely unaddressed. Authors argue overhead is acceptable and provide time + space arguments (≤2× memory, and total time still less than retraining/FATS); but this is mostly “engineering justification,” not a rigorous cost model.

(4) Asked for guidance model performance on forget/remain — Addressed. Authors provide accuracy over training time for guidance vs retraining, and conclude guidance is “ignorant” but not a good final model.

(5) Minor Figure clarity issues — Addressed.

Reviewer 4v3L:


(1) “If you can train an ignorant guidance model, why not just use that / retrain? Training guidance isn’t trivial; practicality unclear; when is it better/worse than retraining?” — Partially addressed (and reviewer appears unconvinced). Authors argue guidance has poor utility so can’t be used directly; they claim guidance training is fast because it only needs “ignorance,” reuse code paths, and adds limited overhead; they also claim FF-Erase is always cheaper in their experiments. Reviewer replies they’re “still not very convinced,” and asks again for conditions where it’s better/worse.

(2) Minor clarity + citation issues. Addressed.

Reviewer y2EG:
(1) “Why not use the original model as guidance (SCRUB-style)?” — Addressed (conceptually). Authors say “moving away” from original goodness can yield invalid goodness distributions and collapse; hence they need an “ignorant” guidance target. (Reviewer follow-up: still leaning accept; limited confidence due to unfamiliarity with FF.)

(2) Reviewer pointed out missing literature. -- Addresed. Authors say they added Appendix A discussion + Appendix C.3 experiments; they argue distillation-based GA fails in FF because it uses final-layer logits and/or creates invalid goodness distributions

**Reviewer Scores:**

I dont believe any of the reviewers would raise their scores. One positive reviewer is unsure. I believe discussions amongst the reviewers after the rebuttal phase would have shifted the sentiment to rejection. Two reviewers remain unconvinced about practicality and necessity. The method adds significant overhead, lacks theoretical grounding for collapse avoidance (or even a mechnaistic understanding/intuition), and depends on a stronger-than-standard verification assumption.

---

### Decision · Program_Chairs · 2026-01-26

Reject